# Tracking the evolution of esophageal squamous cell carcinoma under dynamic immune selection by multi-omics sequencing

Sijia Cui[1,2,7], Nicholas McGranahan[3,7], Jing Gao[4], Peng Chen[4], Wei Jiang[4], Lingrong Yang[5], Li Ma[4], Junfang Liao[4], Tian Xie[6], Congying Xie[1] ✉, Tariq Enver ®[3] ✉ & Shixiu Wu ®[1,4] ✉

Intratumoral heterogeneity (ITH) has been linked to decreased efficacy of clinical treatments. However, although genomic ITH has been characterized in genetic, transcriptomic and epigenetic alterations are hallmarks of esophageal squamous cell carcinoma (ESCC), the extent to which these are heterogeneous in ESCC has not been explored in a unified framework. Further, the extent to which tumor-infiltrated T lymphocytes are directed against cancer cells, but how the immune infiltration acts as a selective force to shape the clonal evolution of ESCC is unclear. In this study, we perform multi-omic sequencing on 186 samples from 36 primary ESCC patients. Through multi-omics analyses, it is discovered that genomic, epigenomic, and transcriptomic ITH are underpinned by ongoing chromosomal instability. Based on the RNA-seq data, we observe diverse levels of immune infiltrate across different tumor sites from the same tumor. We reveal genetic mechanisms of neoantigen evasion under distinct selection pressure from the diverse immune microenvironment. Overall, our work offers an avenue of dissecting the complex contribution of the multi-omics level to the ITH in ESCC and thereby enhances the development of clinical therapy.

Esophageal squamous cell carcinoma (ESCC) is one of the most prevalent cancer types which occurs in Eastern Asia and parts of Africa[1,2]. Several large-scale sequencing studies have revealed the complex genomic landscape of ESCC[3–8]. However, oncological biomarkers might be confounded by sampling bias derived from spatial intratumoral heterogeneity (ITH). ITH, which supplies the fuel for clonal evolution and drug resistance[9], is pervasive across multiple levels of molecular features. A precise understanding of ITH is crucial for the development of effective diagnosis and biomarker design.

Recently, several studies have performed multi-regional whole-exome sequencing (WES), methylation profiling, or TCR sequencing to unveil ITH in ESCC[10–12]. However, ESCC ITH has not been comprehensively characterized across multi-omics from a large cohort of ESCC patients. Thus, an understanding of the complex interplay between the genome, transcriptome, and methylome underpinning ITH in ESCC and the evolution of this disease is lacking. Moreover, the interaction between the cancer cell and its immune microenvironment and how their cross-talk influences cancer evolution has not been investigated. Indeed, although a strong correlation between TCR (T cell receptor) repertoire and genomic ITH has been found in ESCC, mechanisms of neoantigen evasion in ESCC remain unclear.

[1]Department of Radiation and Medical Oncology, Second Affiliated Hospital of Wenzhou Medical University, Wenzhou, China. [2]Department of Medicine, Baylor College of Medicine, Houston, TX, USA. [3]Cancer Institute, University College London, London, UK. [4]National Cancer Center/National Clinical Research Center for Cancer/Cancer Hospital & Shenzhen Hospital, Chinese Academy of Medical Sciences and Peking Union Medical College, Shenzhen, China. [5]Hangzhou Cancer Institute, Hangzhou Cancer Hospital, Hangzhou, China. [6]College of Pharmacy, School of Medicine, Hangzhou Normal University, Hangzhou, China. [7]These authors contributed equally: Sijia Cui, Nicholas McGranahan. ✉e-mail: wzxiecongying@163.com; t.enver@ucl.ac.uk; wushixiu@medmail.com.cn

In this study, we prospectively collected 186 samples from 36 ESCC tumors from 36 patients (Supplementary Fig. S1). All samples (including adjacent normal tissues) were characterized by whole-exome sequencing (WES). We calculated CpG-rich methylation on a genome-wide scale for all the patients by single-cytosine-resolution DNA methylation analysis using reduced representation bisulfite sequencing (RRBS) (Fig. 1a). We also performed RNA sequencing (RNA-seq) for the 33 ESCC patients with high RNA quality. The WES data was derived from each tumor to a median of 260x depth. RRBS data provided more than 10x sequencing coverage (median 77x) on three to six million CpG sites genome-wide for each tumor sample. RNA-seq data achieved a median of 44.5 M high-quality reads. All the detailed information on tri-omics sequencing data is listed in Supplementary Data 1. One main target of our study is to investigate the tumor-intrinsic causes that promote ITH across diverse molecular features. We also explore the hypothesis that intratumor heterogeneity across the diverse molecular features is associated with the selection pressures from distinct tumor microenvironments in ESCC.

## Results

### Genomic ITH in ESCC

Genomic heterogeneity between multi-sites of a single tumor poses major barriers to the biomarker discovery and the development of target therapies[10,13]. To assess the genomic ITH in ESCC, we classified the somatic mutations and copy-number alterations (SCNA) as clonal (prevalent in all cancer cells) or subclonal (present in only a subset of cancer cells). We confirmed early mutational events of somatic mutations in *TP53*, *NOTCH3*, and *PTPRC* in ESCC (Supplementary Fig. S2a).

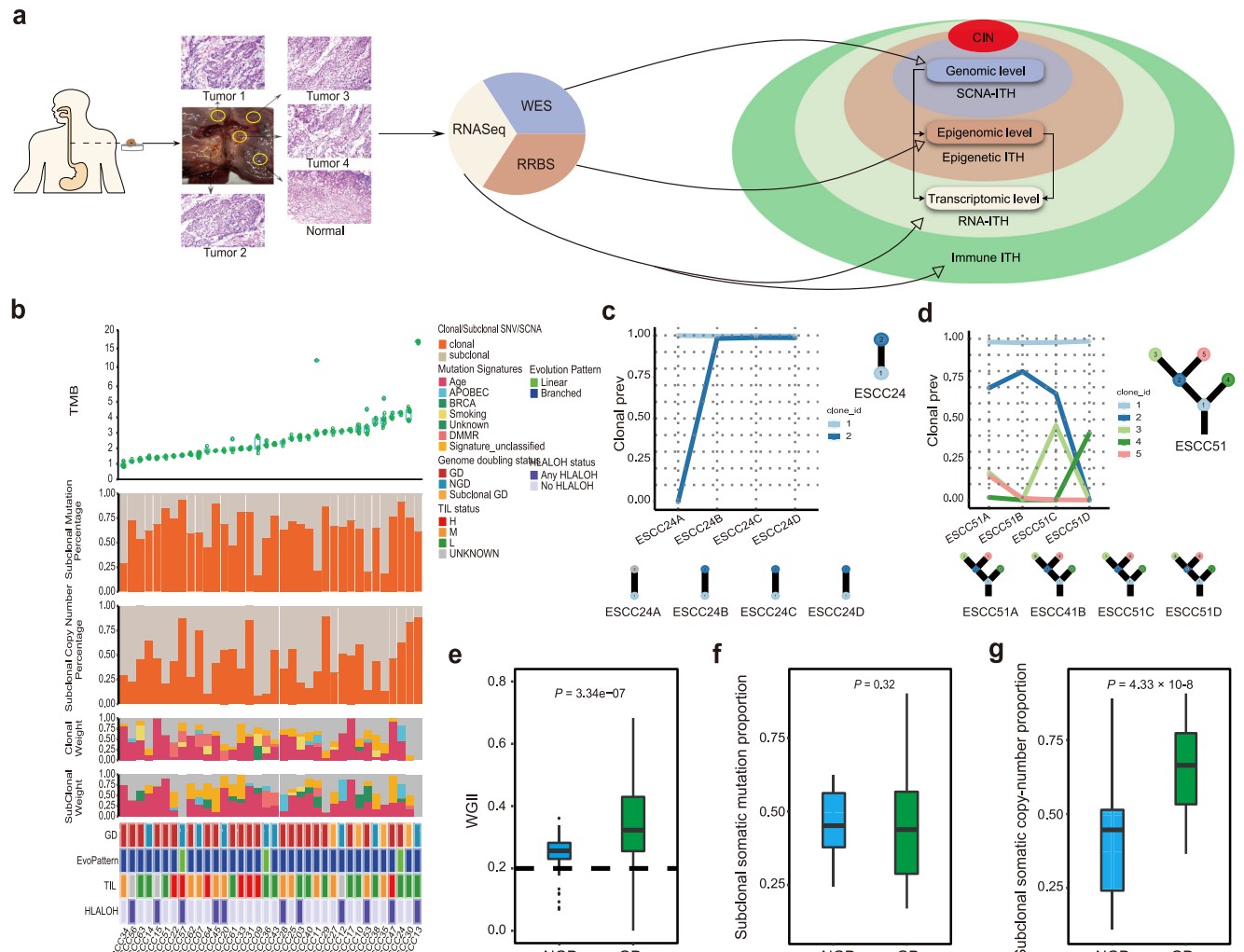

**Fig. 1 | Fuels of Genomic ITH in ESCC. a** Overview of the experimental strategy. Each ESCC multi-sites samples of patients was performed using WES, RNASeq and RRBS to explore the intrinsic multiple omics ITH, the extrinsic TILs ITH and the fuels of each omic ITH. **b** The top panel shows the tumor mutational burden (TMB) for each tumor region (*n* = 186). The minimum and maximum are indicated by the extreme points of the box plot; the median is indicated by the thick horizontal line; and the first and third quartiles are indicated by box edges. The second and third-panel shows the proportion of somatic mutations and copy-number alterations that are identified as clonal (deep blue bars) or subclonal (orange bars) in each patient. The fourth and fifth-panel show the mutation signatures weight identified in the ESCC cohort by using clonal and subclonal somatic mutations separately. The bottom panel shows the summary of the genome doubling (GD), evolution patterns, TIL group, and HLALOH status for each patient (*n* = 36). **c, d** The top panel shows the predicted cellular prevalence of each clonal population for each tumor sample from ESCC24 (**c**) and ESCC51 (**d**) patient; The same color represents the same cell subpopulation across different tumor regions. The bottom panel shows the inferred clonal evolution tree of each tumor sample (the circle in gray means the corresponding clone does not exist in this sample). **e** Box plot represents WGII comparison by genome doubling (GD, *n* = 112) and non-genome doubling (NGD, *n* = 74) status across 186 tumor regions.. **f, g** The box plot shows the subclonal somatic mutation proportion comparison (**f**) and subclonal somatic copy-number proportion comparison (**g**) between GD (*n* = 112) and no GD status (*n* = 74). All Box plot center line represents median value; lower and upper hinges represent 25th and 75th percentiles; the minimum and maximum are indicated by the extremes of the box plot; *P* value is shown; two-sided Wilcoxon rank-sum test.

We observed a median of 33.3% (range 6.1% to 83.4%) of somatic mutations identified as subclonal and a median of 55.1% (range 10.2% to 90.8%) of SCNA as subclonal (Fig. 1b).

Investigating genomic ITH can unveil the evolution history of how mutations are chronologically accumulated. To investigate clone architecture and the evolutionary patterns of each patient, we constructed the tumor phylogeny according to the mutational cellular prevalence (Fig. 1c, d). Each node on the phylogenetic tree represents clonal (shared by all tumor sites) or subclonal (shared by part of tumor sites) mutation clusters and was mapped in the evolutionary history of each ESCC tumor. A total of 207 mutation clusters were identified (Supplementary Fig. S3a, 6 per patient, ranging from 2 to 13). There are 47% of subclones captured in a single branch of the phylogenetic tree (Supplementary Fig. S3b). We also detected a median of 14% (range 0.1 to 53%) of mutations exhibited a clonal illusion[14], whereby they appeared clonal within single samples yet were subclonal in the tumor as a whole,emphasizing the importance of multi-sites sampling (Supplementary Fig. S3c).

The evolutionary trees of our ESCC cohort showed two topological patterns of evolution: the predominant pattern of evolution was the branched pattern (33/36, 92%), and the minor pattern of evolution was the liner pattern 3/36 (8%). This phenomenon was also observed in other cancer types[15,16]. This suggested that ESCC is a much more intratumoral heterogeneous cancer type. For the linear evolutionary pattern, successive clones overgrew their ancestral clones by accumulating somatic alterations without expansion (Fig. 1c and Supplementary Fig. S3a). While the other cases (33/36, 92%) showed a branched pattern, with diverse subclones coexisting and only sharing part of mutations from the ancestral clone (Fig. 1d and Supplementary Fig. S3a). This result revealed the limitation of single sampling for the accurate assessment of genomic ITH. Comparing with the branched evolutionary pattern, we found that linear evolution patterns generally showed limited intratumor heterogeneity, and no clonal expansion occurred during tumor progression. Although the number of patients with linear evolution patterns was limited, we did not observe the purity or number of sites of these patients were significantly lower than the patients with a branched evolutionary pattern. Conceivably, monotherapies against the tumor with the linear evolutionary pattern might show better clinical effects.

## Fuels of genomic ITH in ESCC

**Chromosomal instability.** Ongoing chromosomal instability (CIN) may drive intratumoral heterogeneity[14]. To distinguish CIN+ from CIN tumors, we evaluated the weighted Genome Instability Index (wGII) for each ESCC tumor regions[17]. We observed that higher wGII were found in high-ploidy (ploidy ≥ 3) than diploid tumors (Fig. 1e), which is consistent with previous studies across different tumor types[18]. The allele information showed that genome-doubling events were detected in 70% of ESCC patients and was shared by all the regions except four subclonal genome doubling (GD) tumors (ESCC10, ESCC27, ESCC30, ESCC35), which suggested that GD is frequently an early event during ESCC tumor progression. We observed that there is a significant association between genome doubling and subclonal copy-number alterations (Fig. 1g, $P = 4.33 \times 10^{-8}$, Wilcoxon rank-sum test), but not with regard to mutational heterogeneity (Fig. 1f, $P = 0.32$, Wilcoxon rank-sum test). This suggested that GD events provided the substrates for the genomic ITH at the copy-number alterations level.

Mirrored subclonal allelic imbalance (MSAI), which resulting from different parental alleles being gained or lost in distinct subclones, is another cause of ITH driven by CIN[14,19]. We detected MSAI in 33.3% (12/36) patients and a total of 37 MSAI events from focal to arm level alterations. Besides, this resulted in parallel evolution involving multiple distinct events converging on the same regions in different subclones (Supplementary Fig. S4a–4c).

## Mutagenic processes

To determine which mutagenic processes promote the intratumoral heterogeneity, we systematically analyzed the mutational signatures of both clonal and subclonal stages (Fig. 1b). Using published mutational signatures[20], the number of clonal mutations significantly correlated with the burden of mutations related to aging (Supplementary Fig. S5a, Signature 1a, $P = 0.01$) and DNA mismatch repair (DMMR) processes (Supplementary Fig. S5b, $P = 9.5 \times 10^{-4}$) across the ESCC cohort. We analyzed ESCC13 separately because of the significant burden of signature 15 mutations, which correlated with defective DNA mismatch repair. This might be attributed to the positive MSI status of ESCC13[21]. In comparison to the clonal stage, the contributions of signature 3 (BRAC1/2) significantly correlated with the burden of subclonal mutations (Supplementary Fig. S5d, $P = 2.4 \times 10^{-7}$). Tumors obtained from the 36 ESCC patients got both clonal and subclonal mutations that could also be attributed to DNA mismatch repair signatures (Supplementary Fig. S5e) and aging processes (Supplementary Fig. S5c), which implies that both of the processes continuously induced the mutational events which may contribute to clonal expansion.

## Epigenomic ITH in ESCC

Although informative, the genomic alterations did not fully explain the heterogeneity of ESCC. DNA epigenetic changes also play an important role in the tumorigenesis of ESCC[13]. Firstly, we calculated the methylation level with 1-kb sliding bins across all the tumor regions in our ESCC cohorts. We observed the global hypomethylation trend in ESCC tumor compared with adjacent normal tissue (Fig. 2a and Supplementary Fig. S6a, $P = 4.6 \times 10^{-10}$, Wilcoxon rank-sum test), which was consistent with results from the previous studies[22]. The tumor's hypomethylated bins were significantly enriched in long interspersed nuclear element 1 (LINE-1, L1) and endogenous retrovirus (ERV) regions (Fig. 2c, $P < 0.05$, Fisher's exact test). In contrast, significantly increased DNA methylation levels at CpG islands and promoter regions were detected in ESCC tumors (Fig. 2b, $P < 0.05$, Fisher's exact test). We compared the relative methylation level changes between LINE-1 and LINE-2 regions, which showed that LINE-1 tends to show significantly stronger DNA demethylation than LINE-2 in cancer cells (Supplementary Fig. S6b, $P = 4.43 \times 10^{-13}$, Wilcoxon rank-sum test). L1 is more active and evolutionarily younger than L2[23]. This suggested that the DNA demethylation changes in LINE-1 regions promote the tumorigenesis and progression of tumors, which develop an avenue contrasted to normal tissue development.

To address the epigenetic ITH, we identified the differentially methylated regions (DMR) with the 1-kb sliding bins between the tumor and adjacent normal tissue (Supplementary Fig. S7a). To quantify the methylation level heterogeneity, we calculated the average pairwise ITH (APITH) to measure the methylation ITH of tumors (Methods). The advantage of APITH in ESCC is that its value is not biased by the number of samples of tumor (Supplementary Fig. S6c and Supplementary Fig. S6d). To explore the relationship between the epigenetic ITH and the genomic ITH, we compared the APITH score and the SNV-ITH (proportion of subclonal SNVs). We found a significant correlation between them (Supplementary Fig. S6e, Spearman's rho = 0.49, $P = 0.0037$). This confirmed the conclusion that genetic mutations and epigenetic modifications diversify along with similar patterns during ESCC progression[10].

## Fuels of epigenetic ITH

We next investigate the mechanism underpinning epigenetic ITH. First of all, we calculated the Euclidean distance separately for SCNA profiles and the DNA methylation levels of the DMRs. We observed that the pairwise DNA methylation distances were positively correlated with the pairwise SCNA distances (Fig. 2d, Spearman's rho = 0.63, $P = 4.46 \times 10^{-52}$). To get a whole picture of the interplay between

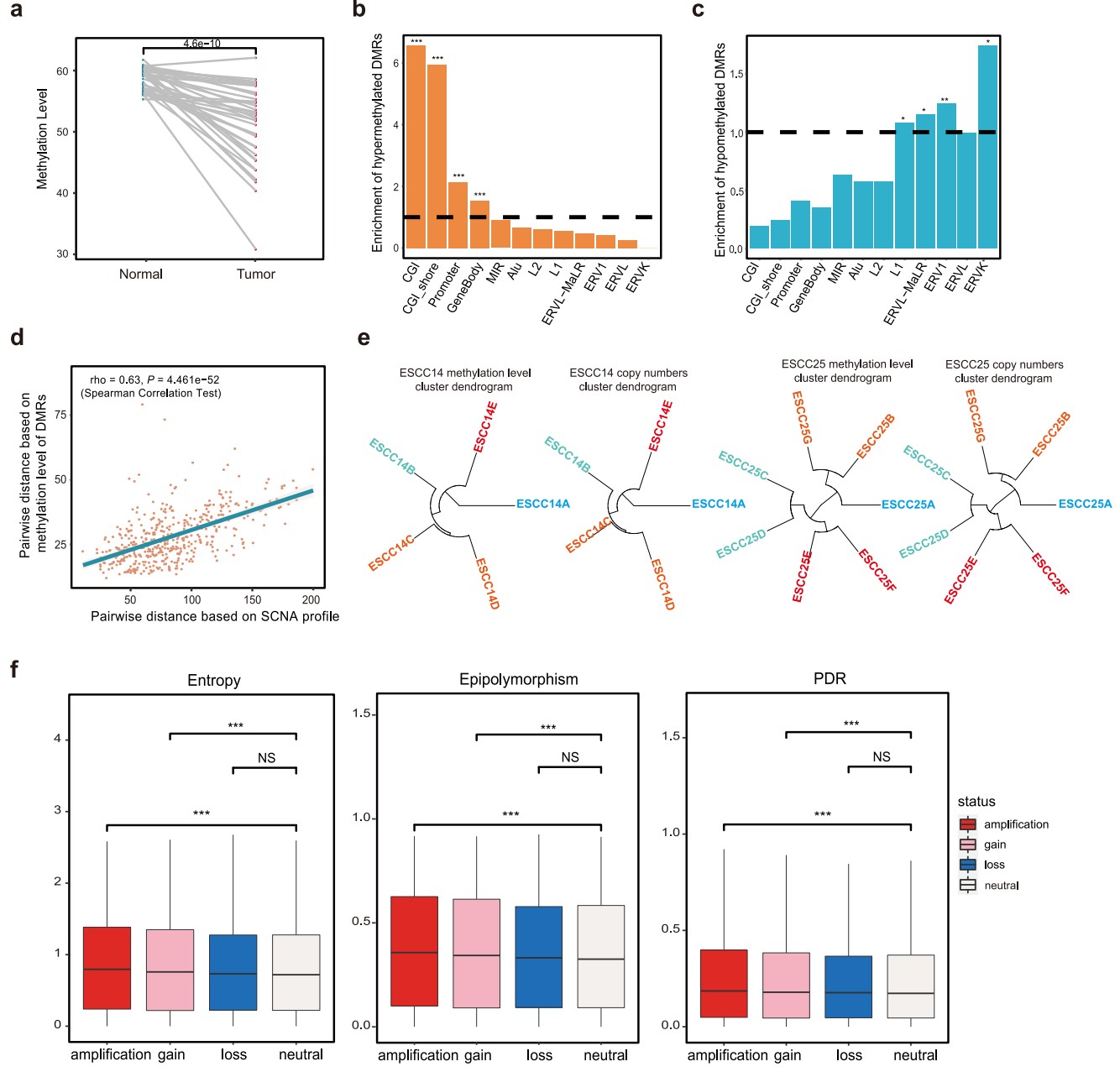

**Fig. 2 | Fuels of epigenetic ITH in ESCC. a** Methylation level comparison between paired Tumor (n = 186) and adjacent normal (n = 36) samples. Two-sided paired t test is used for comparison. **b, c** Enrichment of hypermethylated DMRs (**b**) and hypomethylated DMRs (**c**) for annotated genomic elements between tumor (n = 186) and paired normal (n = 36) samples. *P < 0.05, **P < 0.01, ***P < 0.001; two-sided Fisher's exact test. **d** The pairwise distance of tumor samples from the same patient is based on the DMRs and SCNA profiles. Each dot represents a pair of tumor samples from the same patient. Spearman's correlation coefficient (rho) and corresponding P-value is shown (n = 452 tumor region pairs). Line of best fit shown in blue and gray area represents 95% confidence bands. **e** Phylogenetic and phyloepigenetic trees were constructed using ESCC14 and ESCC25. **f** Distribution of entropy, epipolymorphism, and discordantly methylated read (PDR) scores between neutralnd somatic copy-number alterations regions across 36 ESCC patients with 186 tumor regions. Box plot center line represents median value; lower and upper hinges represent 25th and 75th percentiles; the minimum and maximum are indicated by the extremes of the box plot; ***P < 0.001, NS, no significance, P > 0.05; two-sided Wilcoxon rank-sum test.

genomics and DNA methylation ITH, we exhibited the phylogenetic and phyloepigenetic trees to delineate the relationship among the tumor samples from the same patient. We observed that phylogenetic trees inferred from DNA methylation in several patients, such as ESCC14 and ESCC25, closely recapitulated the phylogenetic trees from SCNA, which is consistent with a previous study[10] (Fig. 2e). Besides, we also find several patients, such as ESCC 17 and ESCC36, got different topologic structures (but with partial similarity) of phylogenetic and phyloepigenetic trees (Supplementary Fig. S7b). This suggested that there is still a potential mechanism that regulates the methylation of

ITH during the evolution of ESCC tumors. (Supplementary Fig. S7b). We further explored the dynamics of chromosomal alterations and the extent to which chromosomal instability may drive epigenetic ITH. DNA-methylation at sequential CpG dinucleotides constitutes a phased epigenetic pattern (epiallele), which provides a snapshot of cells. The diversity of epialleles within the tumor provides a measure of cellular subpopulations within the sample. To further identify the relationship between epigenetic changes and SCNAs, we employed the methclone[24] and epihet[25] to detect the epigenetic loci and the corresponding intratumor methylation heterogeneity which was quantified

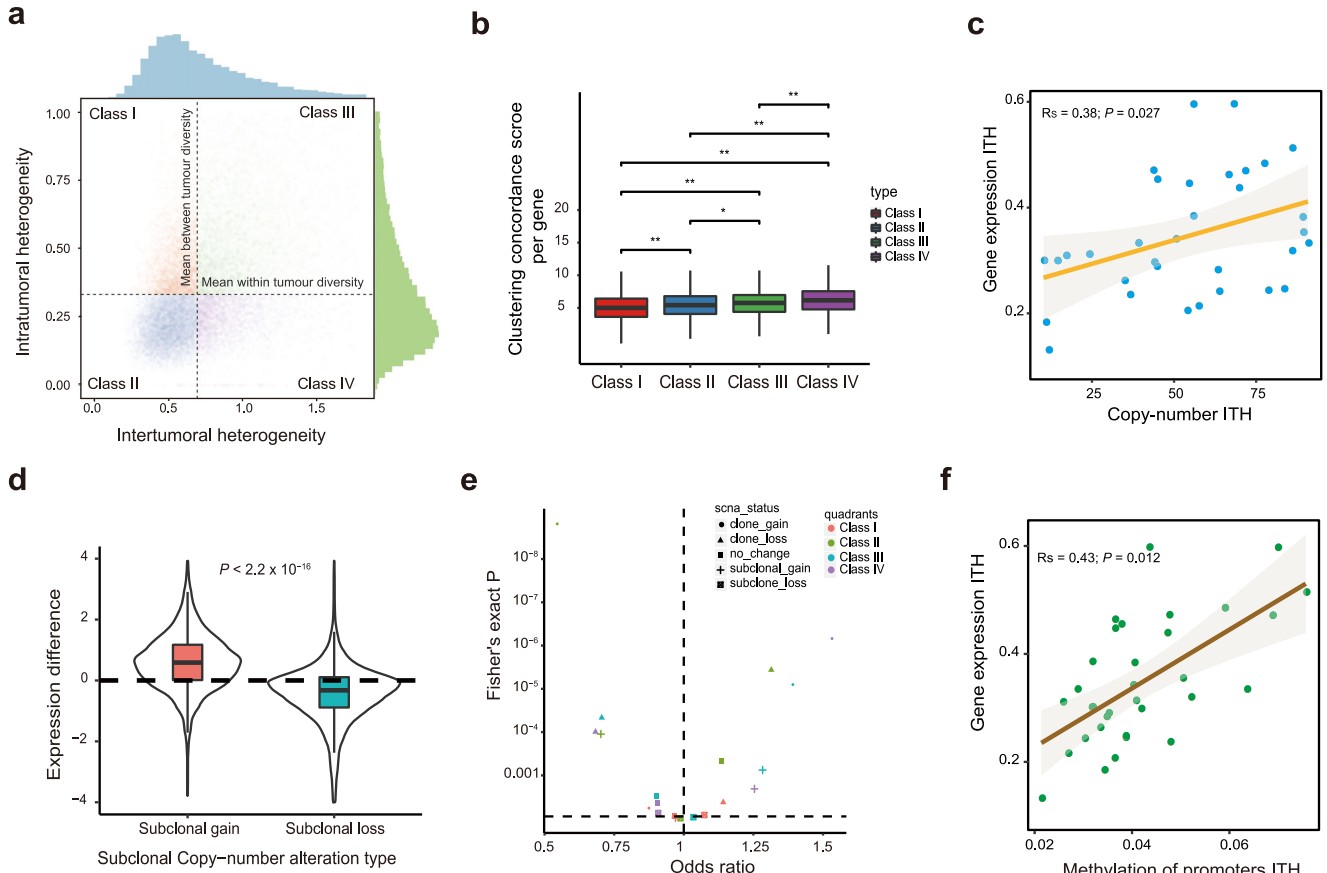

**Fig. 3 | Fuels of transcriptomic ITH in ESCC. a** RNA intertumor (x-axis) and RNA intratumor heterogeneity (y-axis) are shown on the axes using density curves. The genes are split into four classes by the mean intratumor and intertumor heterogeneity scores. The classes are numbered and colored. **b** Boxplots show the clustering concordance score per gene in each class (I = 3354 genes, II = 11622 genes, III = 5368 genes, IV = 3668 genes). The minimum and maximum are indicated by the extremes of the box plot; the median is indicated by the thick horizontal line; and the first and third quartiles are indicated by box edges; *P < 0.05, **P < 0.01, two-sided Wilcoxon rank-sum test. **c** Correlation of gene expression ITH with copy-number ITH. The Spearman correlation coefficient (Rs) between patient-wise RNA-ITH scores and patient-wise SCNA-ITH scores calculated in the ESCC cohort (n = 33 ESCC patients; Rs = 0.38; P = 0.027). Line of best fit shown in orange and gray area represents 95% confidence bands. **d** Violin boxplots show the correlation between subclonal chromosomal copy-number changes and gene expression across 36 ESCC patients. The minimum and maximum are indicated by the extremes of the box plot; the median is indicated by the thick horizontal line; and the first and third quartiles are indicated by box edges; two-sided Wilcoxon rank-sum test. **e** All the genes were classified in clonal/subclonal gain, loss, or no change types. Enrichment was tested by RNA heterogeneity class. Odds ratios are shown using a natural log scale. A two-sided Fisher's exact test was performed. **f** The spearman correlation between patient-wise RNA-ITH and methylation of promoter regions calculated in the 36 ESCC cohort. Spearman's correlation coefficient (Rs) and corresponding P-value are shown. Line of best fit shown in brown and gray area represents 95% confidence bands.

by PDR, entropy, or epipolymorphism. The proportion of discordant reads (PDR), entropy, and epipolymorphism was considered to be a measurement of epigenetic instability. We observed the SCNA regions got an increased entropy level compared with neutral regions. Similarly, SCNA regions with higher entropy also exhibited higher epipolymorphism and PDR levels (Fig. 2f). In our study, we found that CIN provides a higher level of somatic CNA (Fig. 1g). CIN provides more copies of DNA segments, and these segments provide the potential substrate permitting different methylation statuses of CpG sites (Fig. 2f). We could understand this point from the eloci changes. Extra copies will increase the entropy (Epipolymorhism or PDR) of eloci. These results indicated a close relationship between epigenetic variations and CIN.

As LINE-1 is a well-characterized phenomenon in cancer. We further examine the association between the methylation level of LINE-1 and chromosomal instabilities. We correlated the methylation level of L1 and SCNA-ITH, and we found that there is a significant correlation between L1 and SCNA-ITH (Supplementary Fig. S6f, $R^2 = 0.54$, $P = 0.00077$). This suggested that the methylation level of LINE-1 elements is actually associated with CIN in ESCC. Methylation loss in late-

replicating regions makes the heterochromatic structure formation which is called partial methylation domains (PMDs). A recent study showed that PMD demethylation is pervasive in diverse cancer type[26]. To investigate the relationship between epigenetic ITH and CIN, we calculated the PMD APITH scores for each tumor sample and correlated the PMD APITH score with the patient-wise RNA-ITH value. We did not observe a significant correlation between them ($R^2 = 0.32$, $P = 0.11$).

**Transcriptomic ITH in ESCC**
RNA-ITH is a crucial factor that partly was determined by the status of genomic and epigenomic alterations. To identify which genes are more vulnerable to sampling bias in ESCC due to the RNA intratumoral heterogeneity or show higher intertumoral heterogeneity among ESCC patients, we employed a gene classification method[27], which is based on the per-gene metric for RNA intra- and intertumoral heterogeneity. Genes are divided into four RNA heterogeneity quadrants for ESCC (Fig. 3a): low intertumoral heterogeneity and high intratumor heterogeneity (Class I = 3354 genes); both low in intra- and intertumoral heterogeneity (Class II = 11622 genes); both high in intra- and inter-

intertumoral heterogeneity (Class III = 5368 genes); and high intertumoral heterogeneity and low intratumor heterogeneity (Class IV = 3668 genes). Genes in Class IV showed consistent expression within regions and were highly variable between tumors, which provided stronger information for patients' stratification. By using clustering concordance scores of each gene, we observed that Class IV genes outperformed other class types in stratifying ESCC tumor regions by the patient (Fig. 3b). This indicated that genes in Class IV, which prohibited sampling bias and maximizing the difference between patients.

To determine the potential biological features of the four classes, we performed Reactome pathway analysis, respectively (Supplementary Fig. S8a). Class I showed no significant enrichment, RNA splicing pathway enriched in Class II genes, and Class III showed involvement in an extracellular matrix organization. It should be noted that pathways in cell proliferation, including cleavage of the damaged pyrimidine, DNA damage stress-induced senescence, and epigenetic regulation enriched in Class IV, which suggested Class IV genes potentially encode cell proliferation modules.

Moreover, we assessed the RNA-ITH scores based on the number of sequencing regions. We observed RNA-ITH scores increased with the number of sampling regions but became saturated at around four samples for most tumors (Supplementary Fig. S8b). This suggested that at least four regions were required to accurately estimate the RNA-ITH level in ESCC.

## Fuels of transcriptomic ITH in ESCC

We next investigated the potential fuels which contribute to RNA-ITH. Considering the association between CNV and RNA expression, we further explored whether RNA-ITH provided the fuels for RNA-ITH. Overall, we observed a positive correlation between the median RNA-ITH score and SCNA-ITH (Fig. 3c, Spearman's rho = 0.38; $P = 0.027$). The genomic regions with subclonal gained copies also exhibited increased RNA expression, whereas the expression of genes decreased within subclonal lost copies (Fig. 3d). These results suggested that ongoing dynamic chromosomal instability contributes to the ITH of gene expression by altering the copy numbers and dosages of genes. To identify the basic substrates in genomic features for the four quadrants of genes, we performed relative enrichment of clonal and subclonal copy-number changes of genes in different RNA-ITH classes. Class IV genes were enriched in clonal copy-number gain events (odds ratio = 1.59; $P = 1.2 \times 10^{-6}$), which suggested Class IV genes may be driven by clonal DNA copy-number gains selected early in tumor evolution (Fig. 3e).

Epigenetic modifications of the gene's promoter are important for regulating gene expression. To explore the DNA methylation underpinning the RNA-ITH, we detected the different methylated status in the promoter of genes and the RNA expression changes in any paired regions from the same patient. We observed that methylation of first exons and gene promoters related to significant changes in gene expression ($P < 0.001$, Wilcoxon test) (Supplementary Fig. S8c). However, no significant association was observed between methylation of the gene body (excluding promoter regions) and corresponding gene expression ($P = 0.58$, Wilcoxon test). Our results indicate DNA methylation also regulates the RNA-ITH by changing the methylation level of the gene's promoter. We next calculated the ITH of the methylation ITH of genes' promoters and found methylation ITH of gene's promoters positively correlated with the patient-wise RNA ITH (Fig. 3f, Spearman's rho = 0.43; $P = 0.012$). This suggested whole genome epigenetic-ITH contributed more to RNA-ITH compared with SCNA-ITH (Fig. 3c, f). We next measure and rank per gene metric for the methylation intra- and intertumor heterogeneity, and we splited both heterogeneity metrics by their mean value. Similar to the RNA-ITH level, this resulted in four quadrants. Next, we examined the biological significance of the DNA methylation ITH in each quadrant which

correlated with the gene expression ITH in the same quadrant. We performed the Reactome pathway analysis to explore the overlapped genes in each quadrant. We found that Q4 was significantly enriched for pathways involved in cell proliferation and epigenetic regulation. Moreover, we investigate the relationship between RNA-ITH and cellular composition. Overall, we did not observe that RNA-ITH is associated with any immune cell composition and tumor purity (Supplementary Fig. S9a, S9b).

## Immune infiltration drives the heterogeneity across a different omic level

Finally, to determine how immune infiltration varies between and within tumors across the ESCC cohort, we first implemented the published Danaher method[28] to evaluate immune infiltration in the multi-regional ESCC RNA-Seq cohort. Using this method, we profiled RNASeq-determined infiltrating immune cell populations of the 176 tumor regions from 33 ESCC patients, for which the RNA data is qualified.

Unsupervised hierarchical clustering was performed for each immune cell population using the 'ward.D2' method on the Manhattan metric. We revealed two distinct TIL groups, which corresponded to high and low immune infiltration levels (Fig. 4a). Of the 33 ESCC patients with RNA-seq data, 22 patients harbored tumors with consistently high (7 patients (21%)) or low (15 patients (45%)) immune infiltration. Besides, we found that 11 patients (33%) exhibited intratumor heterogeneous immune infiltration (Supplementary Fig. 11c). To further confirm the TILs groups of our ESCC cohort, we next performed Consensus[TME] to estimate the TILs[29], which were positively associated with the Danaher method across different immune cell types (Supplementary Fig. S10a). The distance between pairwise Consensus[TME] distance between every two tumor regions strongly correlated with the pairwise Danaher distance (Supplementary Fig. S10b).

Next, multiplex immunostaining on paraffin sections on the heterogeneous TILs patients for CD4, CD8, CD68, and FOXP3 were taken to validate the difference between high and low immune infiltrated regions from the same patient (Supplementary Fig. S11a). We used the CD4$^+$, CD8$^+$, and FOXP3 T cell density in tumors to quantify the immune infiltration. Consistent with the TILs groups inferred from RNAseq data, tumor regions with high levels of immune infiltration contained higher immune cell density as compared to regions with low levels of immune infiltration (Supplementary Fig. S11b).

Understanding how the immune microenvironment shape tumor evolution may inform strategies to limit tumor adaption in the immune clinical practice[30,31]. We first calculated the Euclidean distance of both somatic mutations and immune infiltration score between any pair of tumor regions from the same tumor (Fig. 4b). We observed a weak correlation between the somatic mutations and the immune microenvironment (Spearman's rho = 0.35, $P = 1.1 \times 10^{-13}$). This result suggested a potential interrelationship between the immune and mutational landscape. Previous evidence suggested that a higher tumor mutation burden (TMB) was likely to harbor more neoantigens as targets for activated immune cells. Tumor with a high mutation burden (>10 mutations per megabase) was found to get a better response in immunotherapy of non-small cell lung cancer[32]. We only observed two ESCC patients (ESCC11 and ESCC13) with at least one tumor region with a high mutation burden (Fig. 1b). Specifically, one tumor region with low immune infiltration of the ESCC11 has a high tumor mutation burden. In contrast, the other tumor regions of ESCC11 with high immune infiltration have a low tumor mutation burden. This result further confirmed the distinct selective pressure on the mutational landscape that existed in different TILs.

Cancer cells can avoid immune-related negative selection through neoantigen depletion or dysfunction of antigen-presenting machinery (APM). We hypothesized that the mechanisms of immune escape might promote genetic clone expansion. The neoantigens from

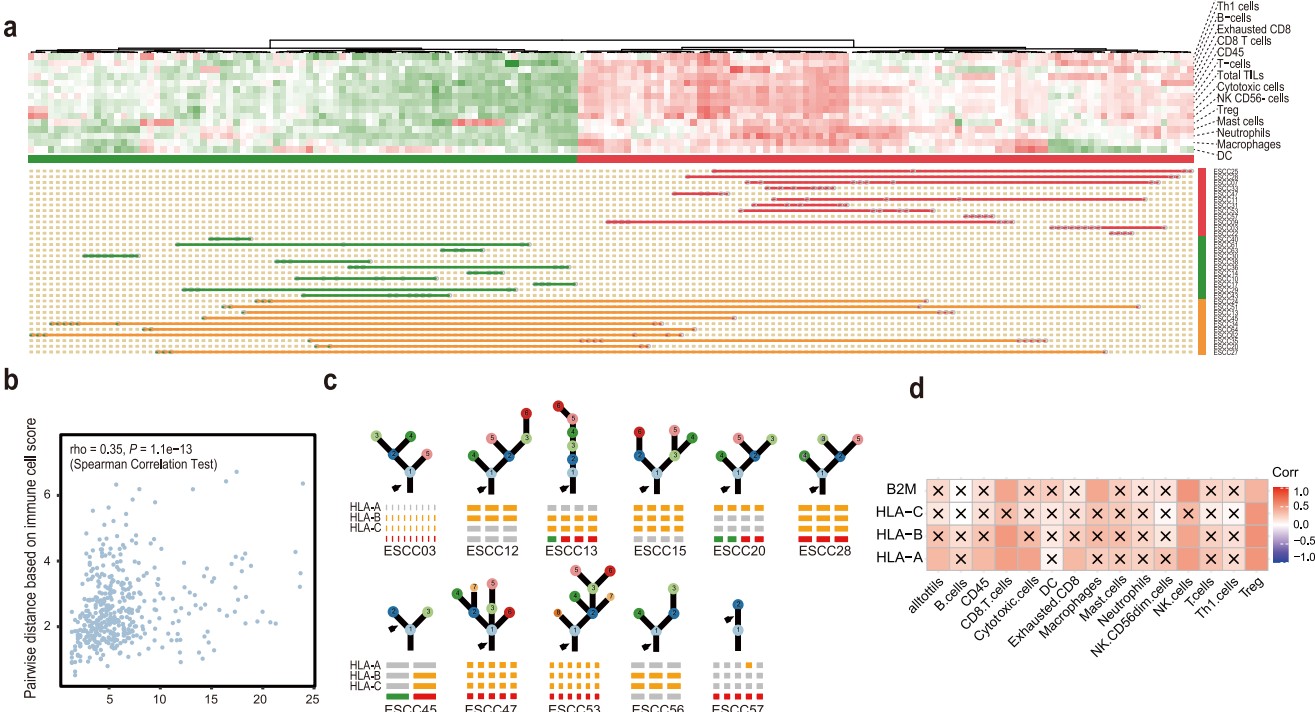

**Fig. 4 | Immune infiltrate heterogeneity in ESCC. a** The ESCC tumor samples ($n = 176$) are clustered by the score of estimated immune infiltrate. The immune cell population score is estimated using the Danaher method. Each row represents a type of immune cell population. Each column represents a tumor region. Tumor regions are classified as having low levels of immune infiltration (low immune) are shown in green; Regions classified as having high levels of immune infiltration (high immune) are shown in red. If all the tumor regions of the same patient are in the high levels of immune infiltration, the patient is marked red. Patients who have tumors that contain heterogeneous levels of immune infiltration are indicated in orange. The patient who has consistently low levels of immune infiltration is indicated in green. **b** Pairwise immune and somatic mutations distances between every two tumor regions from the same patient are compared. Spearman's correlation coefficient (Rs) and corresponding *P*-value is shown. **c** Evidence of HLA LOH has been annotated with the most likely timing of the HLA LOH event ($n = 11$). **d** Correlation heatmap between the ITH gene score of HLA, B2M and ITH score of tumor immune infiltration cells.

nonsynonymous mutations were used to explore neoantigen evasion. By implementing a bioinformatics pipeline to identify neoantigens from the tumours (Methods), a median of 107 predicted neoantigens per tumour (range from 42 to 754) was detected, and neoantigen heterogeneity varied across the ESCC cohort (median 36% neoantigens were found heterogeneously, range from 8 to 89%). In our cohort, we found the neoantigen heterogeneity was significantly higher in heterogeneous immune infiltration (Supplementary Fig. S11f, Wilcoxon rank-sum test, $P = 0.063$), implying the specific selective pressure on the neoantigen from the TILs. We next checked the genetic aberrations of HLA-I, which were the most frequent events leading to APM deficiency. Although HLA mutations could interfere with neoantigen-MHC binding[33], we did not detect non-silent mutations of HLA or B2M in our ESCC cohort. We used the LOHHLA[34] tool to analyze WES data for HLA class I allele loss. Of 36 patients evaluated, we identified nine patients harboring clonal HLA LOH and two with subclonal HLA LOH, suggesting HLA LOH is a prevalent mechanism of APM deficiency in ESCC (Fig. 4c). We found HLA LOH was associated with significantly higher expression of TIL markers, indicating that HLA LOH as an immune evasion mechanism happened under abundant TILs (Supplementary Fig. S11e). We next identified neoantigens, which bind to the lost HLA allele across the ESCC cohort (Supplementary Fig. S11d). We found that mutations predicted to bind to a lost HLA allele exist in all the ESCC patients, which highlights the importance of HLA LOH for immune evasion.

Tumor cells have utilized several means to escape TME recognition through HLA-related mechanisms. They can change the HLA expression to downregulate the expression of MHC complexes and therefore display fewer identifying antigens. The TME is an important factor which affected the HLA expression in ESCC. We assessed the

relationship between HLA gene expression and infiltration of different immune cell types in ESCC. Based on the correlation patterns between immune cell infiltration and HLA and B2M expression, we found that HLA expression showed a significant positive association between HLA and B2M gene expression and the total immune infiltration, CD8 T cells, Cytotoxic cells, and T cells which indicating high HLA gene expression is related with a relatively hot tumor microenvironment (Supplementary Fig. 11g). This result is consistent with previous TCGA research[35]. We further investigate the relationship between the ITH of HLA gene expression and the ITH of TME. We found that the ITH of CD8 T cells and NK cells are significantly correlated with the ITH of HLA-A, HLA-B, and B2M expression (Fig. 4d). This result strongly suggested that the transcriptomic ITH of HLA gene expression is driven by immune infiltration.

## Comparison between Inter-heterogeneity and Intra-heterogeneity in ESCC

The tumor heterogeneity recapitulates tumor heterogeneity in the context of the immune microenvironment in ESCC, but the comparison between inter- or intra-heterogeneity of cancer cell-intrinsic molecular features has not been investigated. To compare the overall intertumoral and intratumoral heterogeneity across multiple omics profile, an integrative unsupervised hierarchical cluster of the multi-regional ESCC cohorts was performed on CNA, gene expression, and DNA methylation data (176 tumor regions; 33 ESCC patients). We revealed there was significant segregation among different patients and perfect clustering concordance between regions from the same tumor, which suggested intertumoral heterogeneity exceeds intratumoral heterogeneity across different omic levels (Supplementary Fig. S12a). To quantify the degree

of within- and between patient heterogeneity, we calculated the Euclidean distance between any tumor sample pairs in the ESCC cohort across the three omics levels. We identified a significant difference between intra-patient and inter-patient tumor pairs in SCNA (Supplementary Fig. S12b, $P < 2.2 \times 10^{-16}$, Wilcoxon rank-sum test), gene expression (Supplementary Fig. S12c, $P < 2.2 \times 10^{-16}$, Wilcoxon rank-sum test), and DNA methylation (Supplementary Fig. S12d, $P < 2.2 \times 10^{-16}$, Wilcoxon rank-sum test). This suggested that each tumor has a uniquely identifiable omics profile on all the copy-number level, gene expression level, and genomic methylation level.

## Discussion

To investigate the potential causes and consequences of ITH in ESCC on different molecular layers, we collected genomic, transcriptomic, and epigenomic data to track how the ITH of tumors are sculpted. We also integrated the multi-omics data to explore how the immune microenvironment shapes the tumor evolution and promotes immune evasion of the tumor.

We observed chromosomal instability might drive the ITH across all the molecular levels. At the genomic level, our results suggested that genome doubling is associated with subclonal copy-number alterations. At the epigenomic level, SCNA regions showed a higher extent of entropy and PDR levels, which suggested CIN may promote epigenetic instability and thereby regulate the epigenetic ITH. At the RNA level, genes with subclonal gain copies also exhibited increased RNA expression or vice versa. This suggests that CIN drives the ITH of gene expression by changing the copy numbers. SCNA contributed directly to the RNA-ITH mainly by dosage effect of copy-number alterations of corresponding genes, and the epigenome regulates the RNA-ITH by methylation level alterations of distant genomic components. To directly identify the immune cell heterogeneity, we calculated the standard deviation of all the immune cell type scores from Consensus$^{TME}$ as a measurement of total immune cell heterogeneity. We next examined the correlation between immune-cell heterogeneity and RNA-ITH. We observed that the immune-cell ITH negatively correlated with RNA-ITH ($R^2 = -0.27$, $P = 0.17$). Although not significant, the result suggested the potential contribution of immune-cell ITH to the RNA-ITH. Considering the character of relatively low purity in ESCC, RNA-ITH does not solely capture cancer cell-intrinsic differences. The immune infiltration could also contribute to the RNA-ITH.

The overall intertumoral heterogeneity significantly outperformed the intratumoral heterogeneity across all the three omics levels, which suggested each tumor showed a uniquely identifiable omics profile. We next profiled the infiltrating immune cell populations of the ESCC cohort using RNA-seq data. We found the immune microenvironment is heterogeneous between and within ESCC patients. 33% of ESCC patients in our cohort showed a distinct status of immune infiltration. The selective pressure from diverse TILs also promoted immune evasion. We found neoantigen heterogeneity showed significantly higher in heterogeneous TILs. HLA LOH is also an immune evasion mechanism of the tumor. We found that 9/36 exhibited clonal HLA LOH and two with subclonal HLA LOH. Besides, we found HLA LOH was associated with significantly higher expression of TIL markers. This suggested HLA LOH is also a potential mechanism of immune evasion in ESCC. We also found that the infiltration of immune cell types was almost uniformly positively associated with HLA and B2M gene expression in ESCC. Besides, the ITH of CD8+ T cells and NK cells are associated with the ITH of HLA and B2M gene expression. This highlights the immune infiltration that drives both genomic and transcriptomic ITH in the ESCC cohort.

Previous lung cancer using multi-regional sequencing across diverse omics studies comprehensively described the spatial and temporal aspects of tumor evolution and the population characteristics of their subclonal evolution[14,27,36,37]. Our results emphasize the importance of both the intrinsic molecular ITH regulation mechanisms

and the selection pressures that the immune system regulates the tumor ITH across different omics levels. Our results suggest that the ITH of ESCC tumors is characterized by multiple independent mechanisms, which involve both inner causes and the outer immune microenvironment. In conclusion, a single tumor site may be insufficient for clinical diagnosis for ESCC patients[38].

## Material and methods

### Patients and sample collection

The cohort in our study was carefully sorted out from 82 ESCC patients who were prospectively collected from Shenzhen Cancer Hospital based on the eligibility criteria. All the patients underwent primary curative resection and received no prior anticancer treatments. The spatially separated tumor specimens were obtained from each individual, with each region at least 0.5 cm away from the others. All the hematoxylin and eosin slides of the ESCC tumor samples were macroscopically reviewed (Tumor purity of each tumor sample was estimated at least 60% by two pathologists, which makes sure all the selected regions were comparable). All samples were immediately frozen in liquid nitrogen and stored at −80 °C. Thin slices of snap-frozen, OCT-embedded tissue blocks were sent for hematoxylin and eosin (H&E) staining. Multi-regional tumors and adjacent non-tumor esophageal tissue from 36 ESCC patients were initially collected for the study. All specimens were collected by surgical resection from Cancer Hospital&Shenzhen Hospital with the written informed consent provided by the patients and the approval by the Research Ethics Committee of Cancer Hospital&Shenzhen Hospital (approval 2019-57). Clinical information (provide in Source Data file) was collected, including gender, age, TNMs stage, smoking and drinking. In total, 186 ESCC tumors and 36 non-cancerous matched normal tissue were performed by the following sequencing experiments. We have consent to publish indirect information including both gender and age. Other clinical information which could identify patients directly are not shown in this study. MOST approval(*BF2022123012755) had been obtained to share genetic data outside of China.

### Multi-regional whole-exome library construction and sequencing

Genomic DNA was taken for whole-exome sequencing. The libraries were constructed by the protocol of Agilent SureSelectXT Human All Exon v6 (Agilent Technologies, CA, USA). First, fragmentation was carried out by the hydrodynamic shearing system (Covaris, USA) to generate 180–280 bp fragments. Next, these short fragments went through a series of library construction steps, such as purification, end blunts, 'A' tailed, adaptor ligation, and amplification. After the PCR reaction, the library hybridizes with the Liquid phase with a biotin-labeled probe, then uses magnetic beads with streptomycin to capture the exons of genes. Captured libraries were enriched in a PCR reaction to add index tags to prepare for hybridization. Products were purified using the AMPure XP system (Beckman Coulter, Beverly, USA) and quantified using the Agilent high sensitivity DNA assay on the Agilent Bioanalyzer 2100 system. After fluorescence quantification by ABI StepOne Plus real-time PCR system (Life technologies), the libraries were sequenced on the Illumina HiSeq X Ten platform, with 150-bp paired-end reads, according to the manufacturer's instructions.

### Sequence alignment and SNV calling

First, paired reads (150 bp) were acquired from Hiseq and evaluated by fastqc (version 0.11.7) (http://www.bioinformatics.babraham.ac.uk/projects/fastqc/). Then, the clean reads were aligned to the reference human genome (build hg19), using the Burrows-Wheeler Aligner (BWA) v0.7.15 to get the alignment file stored in BAM format[39]. SAMTools[40], Picard (http://broadinstitute.github.io/picard/), and GATK (3.6.0)[41] were separately used to sort BAM files, and remove duplicated reads, local realignment, and base quality recalibration to

generate final BAM files. Recalibration training databases included HapMap 3.3, dbSNP build 132, Omni 2.5 M chip, and Mills. Bamdst (https://github.com/shiquan/bamdst) was performed to evaluate the sequence coverage and depth.

Both MuTect[42] (1.1.4) and VarScan2[43] were used to detect SNVs. To identify somatic variants in the target exon data, we first used the Mutect algorithm[42], which detects candidate somatic mutations by the Bayesian statistical analysis of bases and their qualities in both tumor and normal BAM files at a given genomic locus. Variants called by MuTect were filtered according to the filtering parameter 'PASS' Meanwhile, tumor and matched normal pileup files were generated using the "samtools mpileup" command. Then VarScan2 somatic (v2.3.9) utilized the output to identify somatic variants between tumor and matched normal samples. Raw somatic variants were filtered using the VarScan 'processSomatic' subcommand with arguments –min-tumor-freq 0.07, --max-normal-freq 0.02 and –p-value 0.05. An SNV was detected if the mutations were called by both VarScan2 ($p$-value $< = 0.01$) and Mutect when variant allele frequency is greater than 5% and somatic p-value $< = 0.01$.

High-quality somatic variants could be obtained based on the following filtering conditions:

(i) The total reads of the variant site were higher than 10 in both tumor and normal samples.
(ii) VAF < 0.02 and no more than three reads in the normal samples.
(iii) VAF > 0.05 and with more than five reads support variants allele in the tumor samples.

ANNOVAR[44] was utilized to annotate single-nucleotide variants, and SNVs from the 1000 Genome database and dbSNP were removed, but SNVs in the COSMIC database[45] were retained.

Multi-regional WES provides the opportunity to increase the sensitivity to detect low VAF variants[46]. A somatic mutation was not captured by all the tumor regions but only called in part of regions. Read depth and allele information were obtained from the final BAM file using bam-readcount (https://github.com/genome/bam-readcount). Variants were considered to be present if their VAF was more than 0.02 or there were more than three reads supporting the variant allele.

The SNVs from ESCC40A were selected for validation by ultra-deep WES (1200x). A strong relationship was observed between the VAF from the exome-sequencing and the validation sequencing datasets.

We performed multi-regional whole-exome sequencing on 38 ESCC tumors and classified somatic mutations as clonal or subclonal mutations. These mutations were defined as single-nucleotide variants and copy-number alterations, which were shared by all the tumor regions (clonal mutations) or shared by a subset of tumor regions (subclonal)[14]. For the somatic SNV, we merged all the tumor region's SNVs and divided them into two parts: the SNVs shared by all the tumor regions are called clonal SNV, and the SNVs existed in only parts of tumor regions are called subclonal. For the somatic SCNA, we performed bedtools[47] to obtain the SCNA segments shared by all the tumor regions from one patient. The SCNA segments which were shared by parts of tumor regions, were called subclonal SCNA.

## Copy-number analysis

VarScan2 (v2.3.9) was performed to detect the copy-number status from paired tumor and normal samples with default parameters. Chromosomal arm copy-number alteration and the ITH status were detected as the following steps:

(i) Clonal arm gain or loss was detected if all the same chromosomal arms showed at least 75% gain (including amplification) or loss across all the biopsies of the patient. And at least one biopsy showed at least 90% gain or loss.
(ii) Subclonal arm gain or loss was detected if at least one biopsy of the patient showed greater than 75% gain or loss of the chromosomal arm.

Allele-specific copy number and ploidy estimates were generated by using Sequenza[48]. Firstly, we converted VarScan2 output to Sequenza format and further processed it using the Sequenza R package to generate segmented copy-number profile and ploidy estimates. The processed copy-number for each sample was divided by the sample mean ploidy and then $\log_2$ transformed. Gain and loss were defined as $\log_2(2.5/2)$ and $\log_2(1.5/2)$, respectively. Amplification was defined as $\log_2(4/2)$.

To correct purity for the copy number of each segment, we calculate the expected log2ratio for each segment as follows:

$$\log_2\left(\frac{\text{purity} \times \text{cn}_{\text{seg}} + (1 - \text{purity}) \times \text{cn}_n}{\text{ploidy}}\right) \quad (1)$$

The $\text{cn}_{\text{seg}}$, purity, and ploidy are calculated from Sequenza. $\text{cn}_n$ is the copy number of copy neutral region (i.e. 2).

The heterogeneity of CNAs was detected using the minimum overlapped alteration regions. The output generated by Sequenza was subsequently reviewed, and all gene amplifications, homozygous deletions, and loss of heterozygosity were visually inspected using plots of raw $\log_2$ and allele ratios. Genome doubling and wGII for each tumor were determined as previously described[17]. We performed ABSOLUTE to analyze exome sequencing data to identify the genome doubling status of each tumor sample. The weighted genome integrity index (wGII) is calculated by the percentage of SNPs across the genome present at an aberrant copy number, relative to the normal copy number of the tumor sample. The use of percentages helps eliminates the bias induced by differing chromosome sizes. And the wGII score of a sample is defined as the average of this percentage value over the 22 autosomal chromosomes.

## Estimating cancer cell fraction for somatic mutations

To estimate the cancer cell fraction (CCF) for each somatic mutation. The variant allele frequency, copy number in the corresponding region, and the tumor purity were all taken into consideration as previously described[49].

$$\text{CCF} = \text{VAF} \times \frac{1/\text{Purity}}{\text{CN}_{\text{tumor}} \times \text{Purity} + 2 \times (1 - \text{Purity})} \quad (2)$$

CCF is the cancer cell fraction, VAF is the variant allele frequency, $\text{CN}_{\text{tumor}}$ is the copy number at the mutation, and Purity is estimated by Sequenza.

## Subclonal deconstruction

PyClone[50], which implemented the Dirichlet process clustering method, was used to infer the clonal subpopulations based on the mutations and copy numbers. We ran PyClone with 10,000 iterations and a burn-in of 1000 and default parameters. When we constructed the phylogenetic tree based on clonal compositions, two principals were considered: first, the pigeonhole principle, which stated that two mutation clusters could not be considered independently and on separate branches of an evolutionary tree if the sum of the cancer cell prevalence values of the two clusters exceeded 100% within a single tumor region. Second, a descendent clone must exhibit a smaller cellular prevalence than its ancestor within each tumor region, referred to as the "cross rule."

## Detection of MSAI (Mirrored Subclonal Allelic Imbalance)

Mirrored subclonal allelic imbalance discovery involves comparing the BAF of the same heterozygous SNPs across multiple tumor regions and detecting whether the BAF values always follow the same distribution or whether their positions are reversed (mirrored). Germline SNPs were called by using VarScan2, and only SNPs with a minimum coverage of 10× were analyzed. The B allele frequency (BAF) of each SNP was calculated as the ratio of reads of reference base to variant.

Heterozygous SNPs and BAFs were used as input, and mirror subclone allelic imbalances were analyzed by RECUR[19] with default parameters. We also detected a mirrored subclonal allelic imbalance arm gain or loss as parallel evolution events, which is defined as when the opposite parental alleles were affected in at least 75% of a chromosome arm in a minimum of two regions.

## Mutational signature analysis

Both silent and non-silent somatic SNVs were classified as either clonal or subclonal as described, and then the mutational signatures estimation of these SNVs were performed separately by using deconstructSigs[51]. Signatures 1A, 2, 3, 4, 5, 6, 13, 15, 20, and 26 of the published Signatures from Sanger COSMIC[45] were considered. Both clonal and subclonal-specific mutational signature analysis was applied to each spatial biopsies with at least 15 mutations.

**Microsatellite instability.** MSI was estimated using the exome sequence data by MSIsensor[52] version 0.2 with default parameters and filtered using a 0.05 false discovery rate threshold.

## RRBS sequencing

**Library construction and RRBS sequencing.** The RRBS library was constructed using 2 µg of high-quality genomic DNA. Briefly, DNA was restriction digested using the MspI enzyme, which cut the DNA at sites CCGG, then the fragment was end-repaired and dA-tailing to blunt-end products, followed by adaptor-ligation with T overhang. The ligation products were purified by 2% agarose gel electrophoresis and size-selected of DNA fragments 150–400 bp long (including a 100 bp adaptor). Size-selected DNA was bisulfite conversion with the NEXTflex Bisulfite-Seq Kit (Bioo Scientific, Austin, Tx, USA). Bisulfite-converted DNA was then amplified with Illumina PCR primers PE1.0 and 2.0 for 18 cycles. The final library was enriched for fragments with adapters on both ends, and the RRBS was performed by the Illumina HiSeq X Ten platform.

## RRBS quality control and alignment

We preprocessed the raw reads in the fastq format by in-house Perl scripts. Clean reads were obtained by removing reads which contained adapter, poly-N, and low-quality bases in the read end from raw data. Meanwhile, Q20, Q30, and GC content of clean data were evaluated using the fastqc. Then, clean reads were mapped to the reference genome using Bismark alignment software (version 0.18.1)[53]. The mapped reads number, CpG site number, bisulfite conversion rate, and other related details are shown in Supplementary Data 1.

## Bulk purified tumor methylation level from RRBS and Multi-dimensional scaling (MDS)

The bulk tumor methylation level can be explained as the mixture of methylation level in the tumor cells and normal cells. To evaluate the purified tumor cell's methylation level from RRBS data, we calculated the local copy-number state and the purity of sample. The purity were evaluated using the WES dataset, and the local copy number was estimated by HMMcopy. Considering the copy number at each CpG site and the tumor purity, we can calculate the deconvolved tumor methylation rate $m_t$:

$$m_t = \frac{m_b\left(\rho n_t + n_n(1-\rho)\right) - n_n m_n(1-\rho)}{\rho n_t} \qquad (3)$$

The mt should be between 0 and 1, but considering the technical and biological noise, the mt values could be outside of the ranges. We corrected the negative methylation values up to 0 and values greater than one were set to 1.

Next, we aggregated the multi-regional methylation data of each patient. The CpG sites covered by all the samples of each patient were used for further analyses. Then the genome was divided into 3-kb bins (with ≥ 3 CpGs), and the genome-wide DNA methylation level was measured by the mean methylation level of the 3-kb tiles. Then the Euclidean distances between samples were calculated and input to the cmdscale function in R to perform MDS classification. The methylation level of the gene promoter was determined by the mean value of all the methylation sites located in the promoter regions.

## DNA methylation variance among distinct spatial regions

We estimated the region-to-region variance with 3-kb bins. For a given tumor, the standard deviation of methylation levels for a bin across tumor regions was calculated. The variable bins were then ranked with their values. Distribution enrichment of each genomic element in the top 500 variable bins was calculated, and the significance was calculated using Fisher's exact test.

## Identification of DMRs and genomic element enrichment of differentially methylated regions (DMRs)

The CpG sites covered by all the tumor regions and attached normal samples were used for further DMR analysis. Differentially methylated regions (1-kb) were identified by methylKit[54]. Hyper DMRs are the genomic bins with a q-value <0.01 and a percent methylation difference larger than 25%. Hypo DMRs are the genomic bins with q-value <0.01 and percent methylation difference less than −25%. A schema was shown to explain the multi-regional methylation level ITH across the ESCC cohort (Supplementary Fig. 7A). DMRs were annotated on gene promoters (defined as 2-kb upstream and downstream of the transcription start site), first exons, gene bodies, CpG islands, CpG shores (2-kb genomic regions upstream and downstream of CpG islands), and other genomic element regions (using the University of California Santa Cruz (UCSC) table browser) using BedTools[47].

## Quantification of epigenetic ITH

We performed an average pairwise ITH[36] (APITH) to measure the methylation ITH for each patient. The value of APITH is not biased by the number of multi-regional samples per tumor (Supplementary Fig. 6C, D). For each ESCC patient with k tumor regions, we defined $d_{ij}$ as the epigenetic distance between a pair of samples (i, j) based on the methylation level of 1kb-bins, and the APITH is defined as the average across all pairs of samples:

$$\text{APITH} = \frac{2}{k(k-1)} \sum_{1 \le i < j \le k} d_{ij} \qquad (4)$$

## Epiallele shift analysis and epiallele diversity inference

We employed an updated version of methclone[55] (https://github.com/TheJacksonLaboratory/Methclone) to analyze the epiallele composition of each locus. The bam files were provided to the methclone to calculate the dominant methylation pattern information of one locus in the sample. To find suitable thresholds for read coverage, we designed a series of read thresholds set to 40, 60, and 80 reads. We did not find significant differences among the results. So we chose a relatively moderate threshold of 60 reads for methclone to calculate loci. The methclone discards loci that have coverage below 60. To evaluate intratumoral epigenetic heterogeneity, we used an open-source R package epihet[25] to calculate the proportion of discordant reads (PDR), Epipolymorphism, and Shannon entropy.

## Unsupervised clustering analysis

Unsupervised clustering analysis was performed on the copy-number, RNA expression, and methylation level of tumor samples in the ESCC cohort. Briefly, the Euclidean distances were calculated based on the copy number calls, RNA expression of each gene, and the methylation level of each bin in any pair samples. We next integrated all the three

omics distances to perform unsupervised hierarchical clustering. Moreover, the distance of each omics level was also evaluated to compare the concordance within patients and between patients.

## Multi-regional RNA-Seq analysis

**RNA sequencing and alignment.** Total RNA was extracted and purified from fresh frozen tissues using the Trizol reagent (Invitrogen). RNA integrity was measured on an Agilent 2100 Bioanalyzer (Agilent Technologies). Paired samples with high RNA integrity (RNA integrity number > 5), no contaminants, and enough amount of RNA were used to prepare the transcriptome library. mRNA was purified from total RNA using poly-T oligo-attached magnetic beads (Thermo Scientific) and fragmented with an NEB Fragmentation Reagents kit (NEB). The cDNA synthesis, end-repair, A-base addition, and ligation of the Illumina index adapters were performed according to Illumina's TruSeq RNA protocol (Illumina). Library quality was measured on an Agilent 2100 Bioanalyzer for product size and concentration. Paired-end libraries were sequenced by an Illumina HiSeq X Ten ($2 \times 150$-nucleotide read length), with a sequence coverage of 44 M paired reads. FASTQ data underwent quality control and were aligned to the hg19 genome using STAR[56] to the human reference sequence (UCSC hg19 assembly).

## RNA qualification and quantification

Transcript quantification was performed using RSEM with default parameters[57]. And transcript per million (TPM) expression values were generated. Genes were kept with an expression value of at least 1 TPM in at least 20% (35/176) of tumor samples in the multi-region RNA-Seq dataset, and $\log_2(\text{TPM} + 1)$ were used to represent the expression levels. In total, 24,302 genes were applied for further analysis.

## RNA heterogeneity scores

Both Intratumor RNA heterogeneity scores and Intertumor RNA heterogeneity scores were calculated using multi-region RNA-Seq data (normalized count values) from ESCC, which is similar to Dhruva et al. method[27].

## RNA heterogeneity quadrants

We devised the RNA heterogeneity quadrants by splitting the intra- and inter-RNA heterogeneity using their respective mean values.

## Pathway analysis

Pathway enrichment analysis was performed on genes in ESCC Q1-Q4 quadrants using the ReactomePA package (version 1.28.0). Bonferroni-adjusted $P$-value was evaluated, which was based on the threshold $P$-value <0.01.

## Correlated subclonal SCNA with gene expression changes

Based on the heterogeneous copy-number segments between paired tumor regions, we explored the gene expression difference between copy-number alterations and copy-number neutral in the corresponding genes by subtracting the log2 expression value of the SCNA neutral gene from the SCN alteration genes. We then performed a two-sided paired t-test to evaluate the statistical significance.

To examine the enrichment of genes with different copy-number statuses across the four heterogeneity quadrants, we first divided all genes within individual patients into "clonal gain", "clonal loss", "subclonal gain", "subclonal loss" and "no changes" classes. Then we determined whether the different classes of genes across the four quadrants are enriched or depleted by using a two-sided Fisher's exact test.

## Correlation between the differentially methylated promoter and corresponding gene expression changes

We assessed differentially methylated promoters as follows: The methylation levels of all the CpG sites within the promoter of any

paired tumor regions from the same patient were compared. Sites that were differentially methylated at the significant level of 0.05 as determined by Fisher's exact test and had a minimum methylation difference of 0.2 between two tumor regions were considered as differentially methylated promoters. Then we detected the difference between the expression values of corresponding genes in the two tumor regions. Statistical significance was tested with a two-sided paired t-test. Similar steps were also performed in the first exon genomic region.

## Estimating tumor immune infiltration based on RNA-seq

In this part, we performed the Danaher et al. method[28] to evaluate the scores of the immune cell population. The immune signature of this method optimally estimated immune infiltrates compared with other immune cell deconvolution method[37]. The method of Danaher et al. was used to estimate immune cell populations for the tumor region samples by the RNA-seq dataset. The immune cell populations consist of CD4$^+$ T cells (CD4), helper T cells (TH1), regulatory T cells (Treg), dendritic cells (dendritic), B cells (B cell), mast cells (mast), natural killer cells (NK), natural killer CD56$^-$ cells (NK CD56$^-$), neutrophils, macrophages, CD45$^+$ cells (CD45), CD8$^+$ T cells (CD8), exhausted CD8$^+$ T cells (CD8 exhausted), and measures for total T cells (T cells), total TILs (total TIL) and cytotoxic cells (cyto). We also used Consensus$^{\text{TME}}$ method[29] to validate the TILs score of each immune cell population. This method used co-expression patterns in large tumor gene expression datasets to obtain candidate cell type marker genes lists, eliminate numerous false positives and provide high-confidence marker genes. This method provides robust TILs estimation and performs excellently in all cancer-related benchmarks to estimate the existing TILs of tumors. The immune cell population scores of Consensus$^{\text{TME}}$ were normalized to compare with the Danaher method results by using the "ESCA" cancer type. The immune distance was determined by using the Euclidean distance of immune-infiltrate estimates between tumor regions.

## Classifying tumor regions based on the levels of immune infiltration

First, we clustered the samples from the ESCC cohort based on the immune cell population estimated score using "ward.D2". Tumor samples were split into two different groups according to the dendrogram. The samples with higher immune cell population scores were considered to be tumor regions with high levels of immune infiltration. And the samples with lower immune cell populations were considered to be tumor regions with low levels of immune infiltration. If all the tumor regions of the patient were classified as immune high/low, the patient was considered as immune high/low group. If the high immune samples and low immune samples co-existed in the same patient, the overall tumor was classified as a heterogeneous immune group. The three patients (ESCC12, ESCC15, ESCC56) without RNA-seq data were excluded from the immune infiltration analysis.

## Validating tumor immune infiltration using opal

Multiplex staining was performed on 4 mm formalin-fixed paraffin-embedded sections using the Opal multiplex IHC system (PerkinElmer; NEL800001KT) according to the manufacturer's instructions. Briefly, slides were baked for 1 h at 80 °C, followed by deparaffinization with xylene and a graded series of ethanol dilutions (100%, 95%, and 70%), fixation with 10% neutral buffered formalin for 30 minutes, microwave antigen retrieval using the AR9 buffer (PerkinElmer; AR900250ML), and blocking. The antibodies used in this section were CD8 (R&D, MAB1509, clone # 37006, diluted at 1:200), CD56 (R&D, AF2408, Polyclonal, diluted at 1:150), CD4 (Abcam, ab133616, clone EPR6855, diluted at 1:200), CD68 (Abcam, ab213363, clone EPR20545, diluted at 1:200), FOXP3 (Abcam, ab22510, clone mAbcam22510, diluted at 1:200), CK (Abcam, ab756, clone MNF116, diluted at 1:50). To visualize immunofluorescent signaling, the following OPALTM (Perkin Elmer)

TSA dyes were used: OPAL520, 570, 620, 650, 690. Spectral DAPI (Perkin Elmer) was used for the nuclear counterstain. Imaging of slides was performed on a Vectra 3.0 Automated Imaging System (Perkin Elmer). Counterstain was done using DAPI (1:1000) and subsequently mounted using Vectashield (Vectra; H-1000) fluorescence media. Tyrosinase for higher-resolution imaging at 20x magnification using Phenochart (Perkin Elmer). The number of 20x images per sample scanned by the Vectra 3.0 microscope and used for further analysis.

### HLA type and HLA LOH prediction
The HLA type for each patient was detected using POLYSOLVER[33], which uses a normal tissue BAM file as input and employs a Bayesian classifier to determine genotype. Tumour regions with HLA LOH events were detected using LOHHLA[34].

### Neoantigen binders prediction
All 9-11mer peptides that overlapped identified non-silent mutations present in the sample were considered candidate epitopes. MHC-I binding affinity was calculated for every mutation and corresponding wild-type allele using netMHCpan-4.0[58]. The mutant epitope with percentile binding scores of ≤ 2% and equal or better affinity than the wild-type epitope were considered putative neoepitopes. In cases of tumor sample with HLA LOH, predicted neoepitopes associated with the lost HLA allele were excluded (for subclonal HLA LOH, a neoepitope was only excluded if all clones with the neoepitope also exhibited loss of the corresponding HLA allele). When RNA-seq data were available, an expressed neoantigen was detected if at least five RNA-seq reads mapped to the mutation position, and at least three contained the mutated allele.

### Statistical information
We performed all the statistical tests using R. Comparisons of distributions were done using 'wilcox. test' or 't.test'. The two-sided Student's t-test was used to compare the significant differences between the two groups. To consolidate the conclusion of the correlation test of our study, We also performed partial correlation to estimate Pearson (linear) correlation between two variables while controlling for one or more other variables (An R Package for a Fast Calculation to Semi-partial Correlation Coefficients.). This method has been validated in many studies. We computed the partial correlation controlling for tumor purity using the *pcor.test()* function from the R package *ppcor*. All data are presented as mean values ± SEM.

### Reporting summary
Further information on research design is available in the Nature Portfolio Reporting Summary linked to this article.

## Data availability
Raw sequence data (WES, RRBS, and RNASeq dataset) used in the study has been deposited at the European Genome-phenome Archive (EGA), which is hosted by the European Bioinformatics Institute (EBI) under the accession code EGAS00001003832. Please contact the Data Access Committee (DAC) for access to the data: [https://ega-archive.org/dacs/EGAC00001002881]. Data access can be obtained through a request to the corresponding authors. Access to the data will be restricted to non-commercial entities. The corresponding authors will generally respond to requests within five days. Once granted, the access has no time restriction. The remaining data are available within the article, Supplementary Information and Source data provided within this paper. Source data are provided with this paper.

## Code availability
Software code is available at https://github.com/JerrySijia/ESCCMultiregion and can be freely used for educational and research purposes by non-profit institution. For information on the use

for a commercial purpose or by a commercial or for-profit organization, please contact Professor Shixiu Wu.

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

## Acknowledgements

This work was financially supported by the Sanming Project of Medicine in Shenzhen (No. ZSM201812062, No. ZSM201612063), the National Natural Science Foundation of China (No. 81672994). We thank the Research Ethics Committee of Cancer Hospital&Shenzhen Hospital for the ethics.

## Author contributions

Study concepts and design: Shixiu Wu and Tariq Enver. ESCC tissues collection: Peng Chen and Jing Gao. Experimental conduction: Lingrong Yang, Peiyu wu and Jiao Ruidi. Statistical analysis: Sijia Cui, Wei Jiang, Li Ma, Tian Xie and Junfang Liao. Manuscript Writing and editing: Sijia Cui, Nicholas McGranahan and Congying Xie.

## Competing interests

The authors declare no competing interests.
