## [Peer review file · Nature Communications]

REVIEWER COMMENTS

Reviewer #1 (Remarks to the Author): Expert in multi-omics analysis and tumour heterogeneity

Summary

Characterizing intratumoral heterogeneity (ITH) from a multi-omics perspective is crucial to overcome a variety of difficulties in anticancer treatments, ranging from drug resistance to the eventual relapse of a tumor, that arises from the clonal evolution and selection of cancer cells fueled by ITH. In this regard, this study is indeed a valuable resource for the characterization of multi-omics ITH in esophageal squamous cell carcinoma as it provides comprehensive analyses from extensive multi-omics molecular profiles of 186 samples from 36 ESCC patients. It is noteworthy that such profiles have been obtained from multiple regions of a single tumor, so that the direct measurement of ITH is possible by measuring the variability across the multi-region profiles of a single tumor. Using a variety of bioinformatic methodologies, ITH level for each omics level is characterized and correlations between different multi-omics ITH levels were measured. Moreover, the authors argue that the analyses of multi-omics ITH reveals that the evolutionary trajectories of ESCC tumors are shaped by the surrounding immune microenvironment. Overall, despite the novelty of the data analyzed in this study, some of the results are not concrete enough to fully support the argument of the authors, and the manuscript itself needs some polishing to assure its integrity to the readers.

Major Comments

- To support that the selection pressure elicited by the immune microenvironment shapes tumor evolution and promotes the tumor to evade immune responses through the suppression of the expression of neoantigen-producing genes by hypermethylation. I am wondering how the results through Supplementary Fig. S10f-h can be the direct evidence supporting that immune infiltration causes neoantigen silencing by hypermethylation. The scheme of the experiments and comparisons needs to be more clarified. Are the group of genes compared between H and L group is the same in Supplementary Fig 10g and h? If not, the difference between the two groups could have been affected by the 'baseline' methylation levels of the gene, which reflects the characteristics of the group of genes itself, but not the effect of selective pressure on neoantigen silencing. Please specify the detailed experimental scheme for these comparisons so that the readers can follow the arguments.
- I am curious whether we can narrow down the discussion into specific genomic regions that are prone to the variation across subclones so that those regions can be regarded as a major source of multi-omics ITH. For example, we can measure or rank the intra-tumor variance of DNA methylation for each CpG islands or promoters, check if the gene expression ITH of the associated genes are also elevated, and discuss their biological significances.
- In section 'Epigenomic ITH in ESCC', the demethylation of L1 elements was suggested to promote the tumorigenesis and progression of tumors. Since the hypomethylation of retrotransposable elements is already a well-characterized phenomenon in cancer, this argument is indeed valid, but at the same time

it does not involve any novel findings, especially in terms of ITH. I suggest the authors to test whether the increased expressions of L1 elements (measured by RNA-seq data) are associated with the hypomethylation of the corresponding L1 elements, and whether the transposition of L1 elements is actually associated with chromosomal instabilities in ESCC.

- The authors state that no associations were found between abundances of individual immune cell types and RNA-ITH. However, as shown in supplementary figure S8, a large fraction (11/15) of the immune cell types shows nonsignificant, but negative correlation with RNA-ITH, suggesting the possibility of the decreased RNA-ITH due to the predominance of a single immune cell type in tumor microenvironment. Regarding the relatively low tumor purities (ranging from 0.2-0.6) of ESCC samples, I am curious if the RNA-ITH is truly irrelevant of the 'heterogeneity of tumor-infiltrating immune cell types', and I suggest the others to examine the correlation between the immune-cell heterogeneity (such as entropy of immune cell type proportions estimated by CIBERSORT, TIMER and so on) and RNA-ITH, to consolidate that RNA-ITH mainly captures the transcriptomic heterogeneity of cancer cells, not immune cells.

- In section 'Immune infiltration drives the heterogeneity across different omic level' (page 14), methylation levels associated with genes with/without expressed neoantigens were compared to support that the selective forces originating from immune cells induces hypermethylation of neoantigen-producing genes (Supplementary Fig. S10f). The conclusion is interesting enough, but to support the argument, methylation levels associated with a fixed set of genes should be compared between the group with neoantigen mutations and the group without neoantigen mutations within those identical set of genes. That is, if the selective force to silence neoantigen-producing gene really exists, there should be the case where a gene harboring neoantigen-producing mutation is specifically silenced by hypermethylation in one sample, while the same gene without such mutations are still expressed in the other sample. The current result is confounded by the general fact that the expression of gene is negatively correlated with the promoter methylation level. I suggest the authors to show such results to support their arguments.

- In section 'Epigenomic ITH in ESCC', the authors mistakenly stated that in tumors, LINE1 and ERV elements are hypermethylated and CpG islands are hypomethylated. It should be fixed as it can mislead readers to entirely different conclusions.

- The measures of epigenetic heterogeneity used in this study, namely entropy, epipolymorphism and PDR, typically range from 0 to 1. However, the values shown in Fig. 2f seem to range from -1 to 1 without the explanation of the transformation procedures applied to those values. I suggest the authors to clearly explain how those values were transformed.

Minor Comments

- Typos and errors

Abstract, line 1: Esophageal -> esophageal

Page 6, line 3: Fig. 2d does not seem to belong here

Page 13, lines 1 and 6: Figure references Fig. 3f (line 1) and Supplementary Fig. S7c (line 6) seem to be mistakenly switched.

Page 14, line 4: The p-value specified in the text does not match to that in the supplementary Fig. S10f.

Page 14, line 8: The p-value specified in the text does not match to that in the supplementary Fig. S10g.

Page 14, line 8: The p-value specified in the text does not match to that in the supplementary Fig. S10h.

- It was often hard to follow the manuscript since some of the first occurrences of the abbreviations are not fully stated.

Introduction, line 1: ESCC -> Esophageal squamous cell carcinoma

Page 5, line 4 from the bottom: chromosomal instability -> chromosomal instability (CIN)

Page 6, line 1: genome-doubling -> genome-doubling (GD)

Page 7, line 12: to be consistent with LINE-1, ERV -> endogenous retrovirus

Page 8, line 2 from the bottom: PDR -> proportion of discordant reads (PDR)

Page 12, line 5 from the bottom: TMB -> tumor mutational burden (TMB)

- Duplication

In page 16, lines 12-16 are the duplication of the sentences right before them.

Sun Kim, PhD

Director, Bioinformatics Institute

Professor, Department of Computer Science and Engineering

Seoul National University

Seoul, Korea

Reviewer #2 (Remarks to the Author): Expert in esophageal cancer genomics

In this work, the authors performed large-scale ITH genomic and epigenomic sequencing of 186 samples from 36 primary ESCC patients. Specifically, WES, RRBS, and RNA-seq were conducted on matched samples, which by itself is an impressive undertaking. Most of the computational analyses were adequate, rational and revealed important insights. Linking genomic/epigenomic ITH with immune evasion/selection is another valuable part of this work. Overall, this work is a very useful dataset for the ESCC community. However, some of the analyses lack depth and appear to be superficial. A number of concerns were identified. There are also missed opportunities with such comprehensive dataset.

1. For the investigation of the methylation ITH at the DMR level, a representative window showing such ITH would be helpful for the reviewers to understand the context. Similarly, the authors concluded that methylation ITH and genetic ITH significantly correlate with each other, and a display of phylogenetic trees generated from either methylation data or mutational data from a few matched samples would be helpful for the readers.

2. One of the key conclusions of the paper is that “CIN may drive epigenetic instability and thereby affect the methylation level changes, which provided the fuel to the epigenetic ITH”. However, evidence and data supporting this point is weak, correlational and surface (Fig.2d-f). The reviewer still finds it difficult to understand why and how CIN may drive methylation ITH.

3. Sample purity is not well controlled and described. The statement of “Tumor purity of each tumor sample was estimated at least 60% by two pathologists, which makes sure all the selected regions were comparable” is not helpful. Samples should be digitally measured for purity using established methods. For multiple results, purity needs to be taken into account (see below).

4. There is little meaningful change in Fig.2e, albeit statistically significant. Furthermore, it could be driven by sample purity: samples which are less pure will show less change in methylation and SCNA. The authors at a minimum need to correct for purity.

5. Similarly, in Fig.3C, purity needs to be controlled. Again, the correlation is not very robust, and can be biased by purity.

6. Does baseline expression potentially skew the analysis of Fig.3a-b? ie. Genes with very low expression tend to have lower variation either inter- or intra-tumor. If so, this needs to be controlled.

7. A few missed opportunities:

1) How about the ITH at the level of large hypomethylation domains (so called "PMD")? Do they also contribute to RNA ITH?

2) In addition to CIN, is there other sources to drive the methylation ITH? Any particular focal CIN, mutations or pathways which are correlated with methylation ITH?

3) When does immune evasion start to occur? Since the authors have the capability of generating phylogenetic trees using either genetic or methylation data, it would be interesting to determine key immune evasion events (such as mutations/copy number changes in immune related processes, including antigen presentation, HLA, Interferon signaling, TNF pathway, etc)

Page 6: "which suggested that GD if frequently an early event during ESCC tumor progression". Typo: "if" should be "is". Also, incorrect citation of Figure 1D in this sentence.

It was written "The tumor's hypermethylated bins were significantly enriched in long interspersed nuclear element 1 (LINE-1, L1) and ERV regions (Fig. 2b, $P < 0.05$, Fisher's exact test). In contrast, significantly decreased DNA methylation levels at CpG islands and promoter regions were detected in ESCC tumors (Fig. 2c, $P < 0.05$, Fisher's exact test)." Should this be the opposite at least based on the Figure? i.e., the hypermethylation is enriched in promoter and hypo is enriched in LINES??

Some of the figures are hard to read. For example, Fig.4A is impossible to see the labels. Also, there only 4 Main figures and some of the Supplementary Figures (e.g., Figure S10) can be moved to Main figures.

Reviewer #3 (Remarks to the Author): Expert in immunogenomics

Cui and colleagues embarked on a "tour de force" to characterize multiple regions sampled from 36 esophageal squamous cell carcinoma. From a data point of view, it is a great contribute to the scientific community as it provides a wealth of data for other groups to explore. On the other hand, the work is not particularly innovative and the authors should have done a better job in exploring the biological and clinical significance of the reported findings. Analyses are often of correlative nature and fail to provide new knowledge to the community.

Ohter comments:

1. What was tumor percentage of the different cases? Was 10x depth in RRBS enough to reliably determine methylation patterns in tumor cells?

2. When the authors state the following: "We confirmed early mutational events of somatic mutations in TP53, NOTCH3 and PTPRC in ESCC (Supplementary Fig. S2a). We observed a median of 33.3% (range 6.1% to 83.4%) of somatic mutations identified as subclonal and a median of 55.1% (range 10.2% to 90.8%) of SCNA as subclonal (Fig. 1b)." It would be important to mention in the results how this was done.
3. I disagree when the authors state that they identified two main patterns of evolution. The main pattern of evolution was a branched pattern while a linear pattern was only identified in 3 samples.
4. Don't understand why some figures have axis disproportional to the data contained in those (for instance, figure 1E)
5. Although published elsewhere, the concept of RNA-ITH should be introduced when reported in the results section. How to distinguish RNA-ITH caused by different cell composition from differences between tumor cell clones?
6. The authors state that: "We observed a significant correlation between the somatic mutations and the immune microenvironment (Spearman's $\rho = 0.32$, $P = 1.1 \times 10^{-13}$)." I think it is a bold statement when the correlation coefficient is only of 0.32. The P value does not say anything about the strength of the correlation.
7. "The neoantigens from nonsynonymous mutations were used to explore neoantigen evasion. By implementing a bioinformatics pipeline to identify neoantigens from the tumours (Supplementary Methods), a median of 107 predicted neoantigens per tumour (range from 42 to 754) was detected, and also neoantigen heterogeneity varied across the ESCC cohort." Potential neoantigens should be identified making use of the RNA analysis as well, thus, with only expressed mutations.
8. "the methylation level of genes with clonal not expressed neoantigen in the high TIL group is significantly higher than the clonal neoantigen in the low TIL group (Supplementary Fig. S10h, $P = 0.00439$)." Was this specifically encountered in genes carrying putative neoantigens? Meaning, when comparing the total of expressed genes vs non-expressed, I would also expect to see increased methylation in the latter.

Point-by-point response to the reviewers' comments:

Reviewer #1 (Remarks to the Author): Expert in multi-omics analysis and tumour heterogeneity

Summary

Characterizing intratumoral heterogeneity (ITH) from a multi-omics perspective is crucial to overcome a variety of difficulties in anticancer treatments, ranging from drug resistance to the eventual relapse of a tumor, that arises from the clonal evolution and selection of cancer cells fueled by ITH. In this regard, this study is indeed a valuable resource for the characterization of multi-omics ITH in esophageal squamous cell carcinoma as it provides comprehensive analyses from extensive multi-omics molecular profiles of 186 samples from 36 ESCC patients. It is noteworthy that such profiles have been obtained from multiple regions of a single tumor, so that the direct measurement of ITH is possible by measuring the variability across the multi-region profiles of a single tumor. Using a variety of bioinformatic methodologies, ITH level for each omics level is characterized and correlations between different multi-omics ITH levels were measured. Moreover, the authors argue that the analyses of multi-omics ITH reveals that the evolutionary trajectories of ESCC tumors are shaped by the surrounding immune microenvironment. Overall, despite the novelty of the data analyzed in this study, some of the results are not concrete enough to fully support the argument of the authors, and the manuscript itself needs some polishing to assure its integrity to the readers.

Major Comments

1. To support that the selection pressure elicited by the immune microenvironment shapes tumor evolution and promotes the tumor to evade immune responses through the suppression of the expression of neoantigen-producing genes by hypermethylation. I am wondering how the results through Supplementary Fig. S10f-h can be the direct evidence supporting that immune infiltration causes neoantigen silencing by hypermethylation. The scheme of the experiments and comparisons needs to be more clarified. Are the group of genes compared between H and L group is the same in Supplementary Fig 10g and h? If not, the difference between the two groups could have been affected by the 'baseline' methylation levels of the gene, which reflects the characteristics of the group of genes itself, but not the effect of selective pressure on neoantigen silencing. Please specify the detailed experimental scheme for these comparisons so that the readers can follow the arguments.

Response to comment: We thank the reviewer for raising this important concern. The group of genes compared between H and L group is not same. Importantly, after methylation level correct based on purity, we did not observe significant

difference between the two groups ($P > 0.05$). This suggests the difference between the two groups are affected by multiple factors included purity or the ‘baseline’ methylation levels of the genes’ promoter. After carefully consideration, we therefore decided to remove this section of the manuscript and the conclusions which included Supplementary Figure 10F-H. Thanks again for pointing out the problem of our manuscript.

2. I am curious whether we can narrow down the discussion into specific genomic regions that are prone to the variation across subclones so that those regions can be regarded as a major source of multi-omics ITH. For example, we can measure or rank the intra-tumor variance of DNA methylation for each CpG islands or promoters, check if the gene expression ITH of the associated genes are also elevated, and discuss their biological significances.

Response to comment: We appreciate for your suggestions. According to the reviewer’s suggestion, we calculated the epigenetic ITH score for the promoter of each gene (See the revised Methods). We rank the ITH score of DNA methylation for each promoter and then correlated this with the gene-wise expression ITH of the corresponding genes. We selected the most significant genes (total 133 genes, $R^2 > 0.5$, $P < 0.05$) which show higher methylation ITH and higher gene-wise expression ITH. However, when we performed a Reactome pathway analysis to explore the biological significances we did not observe any significant biological pathways. This suggests that the set of genes that show significant correlation between RNA-ITH and DNA ITH do not represent a core cancer pathway. Rather, we observe multiply different pathways being affected.

3. In section ‘Epigenomic ITH in ESCC’, the demethylation of L1 elements was suggested to promote the tumorigenesis and progression of tumors. Since the hypomethylation of retrotransposable elements is already a well-characterized phenomenon in cancer, this argument is indeed valid, but at the same time it does not involve any novel findings, especially in terms of ITH. I suggest the authors to test whether the increased expressions of L1 elements (measured by RNA-seq data) are associated with the hypomethylation of the corresponding L1 elements, and whether the transposition of L1 elements is actually associated with chromosomal instabilities in ESCC.

Response to comment: Thank you for the important suggestion. To quantify the LINE-1 expression levels, similar to previous study¹, we performed BWA to aligned RNA-seq reads against the LINE-1 sequence. Then we normalized the mapped read counts by the total number of aligned RNA-seq reads. We excluded the repeat region (Alu and SVA) from the LINE-1 region. According to the reviewer’s suggestion, we next tested the relationship between the methylation level of L1 elements and the RNA expression of L1 elements. We did not identify any significant relationship between the methylation level of L1 elements and its RNA expression ($R^2 = 0.11$, $P = 0.6$). To examine the association between transposition

of L1 elements and chromosomal instabilities, we correlated the methylation level of L1 and SCNA-ITH. We observed that there is significant correlation between L1 and SCNA-ITH ($R^2 = 0.54$, $P = 0.00077$). This suggested the transposition of L1 elements is actually associated with CIN in ESCC.

4. The authors state that no associations were found between abundances of individual immune cell types and RNA-ITH. However, as shown in supplementary figure S8, a large fraction (11/15) of the immune cell types shows nonsignificant, but negative correlation with RNA-ITH, suggesting the possibility of the decreased RNA-ITH due to the predominance of a single immune cell type in tumor microenvironment. Regarding the relatively low tumor purities (ranging from 0.2-0.6) of ESCC samples, I am curious if the RNA-ITH is truly irrelevant of the 'heterogeneity of tumor-infiltrating immune cell types', and I suggest the others to examine the correlation between the immune-cell heterogeneity (such as entropy of immune cell type proportions estimated by CIBERSORT, TIMER and so on) and RNA-ITH, to consolidate

that RNA-ITH mainly captures the transcriptomic heterogeneity of cancer cells, not immune cells.

Response to comment: We appreciate this important suggestion. First, we must emphasize that the reason for adopting the Danaher approach was that in previous studies exploring other transcriptomic approaches to deconvolve the immune microenvironment, the Danaher immune signature performed best to estimated immune infiltrates and satisfied two important criteria: (1) negatively correlated with tumor purity. (2) not showing a positive correlation with tumor copy number at the gene locus. Nevertheless, to further validate the conclusion, we further use the Consensus^{TME2} to confirm the results. This method compiles cell type-specific genes used by seven published genes or existing TME cell estimation methods which include Bindea *et al.* gene sets³, Davoli *et al.* gene sets⁴, Danaher *et al.* gene sets⁵, CIBERSORT⁶, MCP-counter⁷, TIMER⁸ and xCell⁹ and performed pan-cancer benchmarks using public data from TCGA.

To directly identify the immune-cell heterogeneity, we calculated the standard deviation of all the immune cell type scores from Consensus^{TME} as a measurement of total immune cell heterogeneity. We next examined the correlation between the immune-cell heterogeneity and RNA-ITH. We observed that the immune-cell ITH negatively correlated with RNA-ITH ($R^2 = -0.27$, $P = 0.17$). Although not significant, the result suggested the potential contribution from immune-cell ITH to the RNA-ITH.

We could also infer that the immune-cell heterogeneity contributed to the RNA-ITH. This results suggested that RNA-ITH mainly capture the RNA heterogeneity of cancer cell but could also be influenced by non-cancer cell differences. We have therefore added the following to the discussion.

Nevertheless, we agree with the reviewer that unlike mutational ITH, RNA-ITH will also be influenced by non-cancer cell differences. We have therefore added the following to the discussion:

Considering the character of relatively low purity in ESCC, RNA-ITH does not solely capture cancer cell intrinsic differences. The immune infiltration could also contributed to the RNA-ITH.

5. In section ‘Immune infiltration drives the heterogeneity across different omic level’ (page 14), methylation levels associated with genes with/without expressed neoantigens were compared to support that the selective forces originating from immune cells induces hypermethylation of neoantigen-producing genes (Supplementary Fig. S10f). The conclusion is interesting enough, but to support the argument, methylation levels associated with a fixed

set of genes should be compared between the group with neoantigen mutations and the group without neoantigen mutations within those identical set of genes. That is, if the selective force to silence neoantigen-producing gene really exists, there should be the case where a gene harboring neoantigen-producing mutation is specifically silenced by hypermethylation in one sample, while the same gene without such mutations are still expressed in the other sample. The current result is confounded by the general fact that the expression of gene is negatively correlated with the promoter methylation level. I suggest the authors to show such results to support their arguments.

Response to comment: We appreciated for your important suggestion. Considering the purity could affect the conclusion of methylation, we have corrected the methylation level of each methylated sites. Indeed, the two groups indeed are influenced by the ‘baseline’ methylation levels of gene and we found we were too haste to make conclusion. So we decided to deleted this controversial part which included Supplementary Figure 10F-H.

6. In section ‘Epigenomic ITH in ESCC’, the authors mistakenly stated that in tumors, LINE1 and ERV elements are hypermethylated and CpG islands are hypomethylated. It should be fixed as it can mislead readers to entirely different conclusions.

Response to comment: We do appreciate for your careful reading. We have revised the manuscript and further examined all the manuscript to make sure the conclusions keep consistent with the description of figures.

7. The measures of epigenetic heterogeneity used in this study, namely entropy, epipolymorphism and PDR, typically range from 0 to 1. However, the values shown in Fig. 2f seem to range from -1 to 1 without the explanation of the transformation procedures applied to those values. I suggest the authors to clearly explain how those values were transformed.

Response to comment: Thank you for the careful reading and comments. We have double checked the code and we discovered an unfortunate bug in our epiallele pipeline. The measurements of epigenetic heterogeneity are quantified by PDR, entropy, or epipolymorphism. The PDR score was calculated as the proportion of discordant reads, containing both methylated and unmethylated CpGs within a local region that covered at least four CpGs. Epipolymorphism is a measurement of the observed consistency of a given methylation pattern within a region versus the expected random pattern. Obviously, elevated overall epipolymorphism values reflecting higher heterogeneity. In the revised version manuscript, we used the

methclone¹⁰ and Epihet¹¹ to recalculate the entropy, epipolymorphism and PDR of all the loci in the ESCC cohort. We could also observed that regions with high epiallele entropy and high epipolymorphism also exhibited higher PDR values. We have updated both of the manuscript and the figures.

Minor Comments

- Typos and errors
- Abstract, line 1: Esophageal -> esophageal
- Page 6, line 3: Fig. 2d does not seem to belong here
- Page 13, lines 1 and 6: Figure references Fig. 3f (line 1) and Supplementary Fig. S7c (line 6) seem to be mistakenly switched.
- Page 14, line 4: The p-value specified in the text does not match to that in the supplementary Fig. S10f.
- Page 14, line 8: The p-value specified in the text does not match to that in the supplementary Fig. S10g.
- Page 14, line 8: The p-value specified in the text does not match to that in the supplementary Fig. S10h.
- It was often hard to follow the manuscript since some of the first occurrences of the abbreviations are not fully stated.
- Introduction, line 1: ESCC -> Esophageal squamous cell carcinoma
- Page 5, line 4 from the bottom: chromosomal instability -> chromosomal instability (CIN)
- Page 6, line 1: genome-doubling -> genome-doubling (GD)
- Page 7, line12: to be consistent with LINE-1, ERV -> endogeneous retrovirus

- Page 8, line 2 from the bottom: PDR -> proportion of discordant reads (PDR)
- Page 12, line 5 from the bottom: TMB -> tumor mutational burden (TMB)
- Duplication
- In page 16, lines 12-16 are the duplication of the sentences right before them.

Response to comment: We do appreciate for your careful reading and apologize for the writing. In the revised manuscript, all the typos and errors listed are updated, and we hope the revised version could satisfy the reviewer criticisms.

Reviewer #2 (Comments to the Author)

In this work, the authors performed large-scale ITH genomic and epigenomic sequencing of 186 samples from 36 primary ESCC patients. Specifically, WES, RRBS, and RNA-seq were conducted on matched samples, which by itself is an impressive undertaking. Most of the computational analyses were adequate, rational and revealed importantly insights. Linking genomic/epigenomic ITH with immune evasion/selection is another valuable part of this work. Overall, this work is a very useful dataset for the ESCC community. However, some of the analyses lack depth and appear to be superficial. A number of concerns were identified. There are also missed opportunities with such comprehensive dataset.

1. For the investigation of the methylation ITH at the DMR level, a representative window showing such ITH would be helpful for the reviewers to understand the context. Similarly, the authors concluded that methylation ITH and genetic ITH significantly correlate with each other, and a display of phylogenetic trees generated from either methylation data or mutational data from a few matched samples would be helpful for the readers.

Response to comment: We appreciate for your comments. To illustrate the correlation between the methylation ITH and copy number alteration ITH, we implemented the phylogenetic trees from both methylation and copy-number alteration data of several patients. It should be noted that the methylation level is corrected by purity. For each patient, pairwise Euclidean distances were calculated between all tumor sites using copy number and methylation level of each genomic bins (See Methods). Copy number phylogenetic trees and phyloepigenetic trees were both constructed from these pairwise distances. We observed that the phylogenetic trees inferred from DNA methylation closely recapitulated phylogenetic trees from copy-number alterations. This results further consolidate the significant correlation between methylation ITH and genetic ITH.

2. One of the key conclusions of the paper is that “CIN may drive epigenetic instability and thereby affect the methylation level changes, which provided the fuel to the epigenetic ITH”. However, evidence and data supporting this point is weak, correlational and surface (Fig.2d-f). The reviewer still finds it difficult to understand why and how CIN may drive methylation ITH.

Response to comment: I do appreciate for your comments. I agree with the reviewer that the conclusion is aggressive. We have updated the statement “CIN may drive epigenetic instability and thereby affect the methylation level changes, which provided the fuel to the epigenetic ITH” in the manuscript. In our study, we found that CIN provide a higher level of subclonal somatic CNA (Figure 1G). And the methylation level of CpG sites significantly change in the subclonal somatic CNA regions. Besides, we could also observe that the CIN change the methylation level of corresponding regions. Also if the copy number alterations are shared by parts of tumor regions, the methylation level located in the subclonal SCNA segment will also change. CIN provide more copies of DNA segments and these segments provide the potential substrate permitting different methylation status of CpG sites (Figure 2D). We could understand this point from the eloci changes. Extra copies will increase the entropy (Epipolyorphism or PDR) of eloci. These results indicated a close relationship between epigenetic variations and CIN.

3. Sample purity is not well controlled and described. The statement of “Tumor purity of each tumor sample was estimated at least 60% by two pathologists, which makes sure all the selected regions were comparable” is not helpful. Samples should be digitally measured for purity using established methods. For multiples results, purity needs to be taken into account (see below)

Response to comment: We do appreciate for your careful reading and comments. The purity of each sample has been estimated using Sequenza¹² and the results were displayed in Supplementary Figure S8B. We have corrected the methylation level and SCNA profile results upon the purity.

4. There is little meaningful change in Fig.2e, albeit statistically significant. Furthermore, it could be driven by sample purity: samples which are less pure will show less change in methylation and SCNA. The authors at a minimum need to correct for purity.

Response to comment: I appreciated for the comments. According to the reviewer’s suggestion, we have validated the conclusion. To avoid the purity influence, we chose different status SCNA regions from one tumor sample. We simultaneously calculated the Entropy, PDR and Epipolymorphism of eloci in the corresponding SCNA regions. We could observe that from the same tumor sample, eloci within higher copy number alterations regions showed increased entropy level. Besides, SCNA regions with higher entropy exhibited higher PDR and epipolymorphism values. These results confirmed our conclusion. We also agree with the suggestions from the reviewer about the purity and we have updated the figures which use result from single sample.

5. Similarly, in Fig.3C, purity needs to be controlled. Again, the correlation is not very robust, and can be biased by purity.

Response to comment: I appreciated for the comments. According to the reviewer suggestion, we employed a multiple regression model to correct purity. The result improved a little. Considering we have examined the relationship between RNA-ITH and tumor cellular composition in the ESCC patients, we observed that RNA-ITH did not correlated with any of the immune cell subsets. These results are consistent with previous study¹³.

6. Does baseline expression potentially skew the analysis of Fig.3a-b? ie. Genes with very low expression tend to have lower variation either inter- or intra-tumor. If so, this needs to be controlled.

Response to comment: We do appreciate your comments. First, we have applied an expression filter to keep genes with an expression value of at least 20% of tumor samples in the ESCC multi-regional RNA-seq dataset to avoid the genes with very low expression across all the samples in our cohorts. Besides, genes with low expression tend to be lower variation will be classified into Class II quadrant. In our study, we mainly focused on the variation of RNA expression between different regions or among patients. Besides, this also could help us to explore the causes of the genes with very low expression, which are possibly driven by genomic copy number loss or hypermethylated promoter of corresponding genes¹⁴.

7. A few missed opportunities:

- 1) How about the ITH at the level of large hypomethylation domains (so called “PMD”)? Do they also contribute to RNA ITH?

Response to comment: We appreciated for the reviewer’s suggestions. We calculated the PMD APITH scores and correlated the PMD APITH score with patient-wise RNA ITH value. We did not observe significant correlation between them ($R^2 = 0.32$, $P = 0.11$).

We added these important results to the manuscript:

Methylation loss in late-replicating regions makes the heterochromatic structure formation which is called partial methylation domains (PMDs). Recent study showed PMD demethylation is a pervasive in diverse cancer type¹⁵. We calculated the PMD APITH scores for each tumor samples and correlated the PMD APITH score with patient-wise RNA-ITH value. We did not observe significant correlation between them ($R^2 = 0.32$, $P = 0.11$).

2) In addition to CIN, is there other sources to drive the methylation ITH? Any particular focal CIN, mutations or pathways which are correlated with methylation ITH?

Response to comment: Thank you for the comments. After carefully correlation analysis, unfortunately, we did not detect any other sources to drive the methylation ITH. We will explore the question in the future study.

3) When does immune evasion start to occur? Since the authors have the capability of generating phylogenetic trees using either genetic or methylation data, it would be interesting to determine key immune evasion events (such as mutations/copy number changes in immune related processes, including antigen presentation, HLA, Interferon signaling, TNF pathway, etc)

Response to comment: Thanks for the reviewer's suggestions. HLA LOH event is an important immune evasion mechanism in ESCC. To delineate the time of HLA LOH event, we have annotated the event on the phylogenetic trees (Figure 4D). And it should be noted that HLA LOH is mostly early event during the ESCC tumor evolution.

4) Page 6: "which suggested that GD if frequently an early event during ESCC tumor progression". Typo: "if" should be "is". Also, incorrect citation of Figure 1D in this sentence.

It was written "The tumor's hypermethylated bins were significantly enriched in long interspersed nuclear element 1 (LINE-1, L1) and ERV regions (Fig. 2b, $P <$

0.05, Fisher's exact test). In contrast, significantly decreased DNA methylation levels at CpG islands and promoter regions were detected in ESCC tumors (Fig. 2c, $P < 0.05$, Fisher's exact test)." Should this be the opposite at least based on the Figure? i.e., the hypermethylation is enriched in promoter and hypo is enriched in LINEs??

Some of the figures are hard to read. For example, Fig.4A is impossible to see the labels. Also, there only 4 Main figures and some of the Supplementary Figures (e.g., Figure S10) can be moved to Main figures.

Response to comment: We do appreciate for your careful reading and apologize for the writing. In the revised manuscript, all the typos and errors listed are updated, and we hope the revised version could satisfy the reviewer criticisms.

The conclusion is obviously and entirely different. We have revised the manuscript and further examined all the manuscript to make sure the conclusions keep consistent with the description of figures.

Reviewer #3 (Comments to the Author)

1. What was tumor percentage of the different cases? Was 10x depth in RRBS enough to reliably determine methylation patterns in tumor cells?

Response to comment: We do appreciate for your comments. The estimated tumor percentage was shown in Supplementary Figure S8B. The 10x depth in RRBS was one of the quality control measurements to calculate the methylation levels of each CpG sites. We calculated the methylation pattern from the same location defined by four adjacent CpGs covered by the same read by methclone¹⁰. To find suitable thresholds for read coverage, we designed a series of read thresholds set to 40, 60 and 80 reads. We did not find significant differences among the results. So we chose a relatively moderate threshold of 60 reads for methclone to calculate a loci which is consistent with previous study^{10, 16}.

2. When the authors state the following: "We confirmed early mutational events of somatic mutations in TP53, NOTCH3 and PTPRC in ESCC (Supplementary Fig. S2a). We observed a median of 33.3% (range 6.1% to 83.4%) of somatic mutations identified as subclonal and a median of 55.1% (range 10.2% to 90.8%) of SCNA as subclonal (Fig. 1b)." It would be important to mention in the results how this was done.

Response to comment: Thank you for the comments. We performed multiregion whole-exome sequencing on 38 ESCC tumors and classified somatic mutations as clonal or subclonal mutations. These mutations were defined as single-nucleotide variants, and copy-number alterations, which were shared by all the tumor regions (clonal mutations) or shared by a subset of tumor regions (subclonal)¹⁷. For the somatic SNV, we merged all the tumor regions SNVs and divided them into two parts: the SNVs shared by all the tumor regions are called clonal SNV and the SNVs existed in only parts of tumor regions are called subclonal. For the somatic SCNA, we performed bedtools¹⁸ to obtain the SCNA segments shared by all the tumor regions from one patient. The SCNA segments which were shared by parts of tumor regions were called subclonal SCNA. And in the revised manuscript, we have provided this information to the method part of our manuscript as suggested to make it more precise.

3. I disagree when the authors state that they identified two main patterns of evolution. The main pattern of evolution was a branched pattern while a linear pattern was only identified in 3 samples.

Response to comment: We appreciate for your comments. According to your suggestion, we have updated the statement. In our study, we observed two topological patterns of evolution: the predominant pattern of evolution was branched pattern and the minor pattern of evolution was linear pattern. This phenomenon also observed in other cancer types^{19,20}. We think this might be attributed to the ITH level in different cancer types. ESCC is a much more intratumoral heterogeneous cancer type and the frequency of linear pattern is relatively lower.

4. Don't understand why some figures have axis disproportional to the data contained in those (for instance, figure 1E)

Response to comment: We appreciate for your reminding. When we combined the figures using the adobe illustrator, the width and height were not proportionally resized and the numbers look squished. We have updated the figures and make sure the proportionally resized figures were well arranged in the panel.

5. Although published elsewhere, the concept of RNA-ITH should be introduced when reported in the results section. How to distinguish RNA-ITH caused by different cell composition from differences between tumor cell clones?

Response to comment: We appreciate for your reminding. According to your suggestion, we have added the RNA-ITH description in the ‘Transcriptomic ITH in ESCC’ part of our manuscript. Transcriptomic intratumor heterogeneity (RNA-ITH) is used to measure the variation of gene expression patterns between different regions of the individual tumor²¹. RNA-ITH confounded existing expression-based biomarkers in ESCC. We examined the relationship between RNA-ITH and tumor cellular composition in our ESCC cohort. RNA-ITH did not significantly correlated with any of the immune cell type inferred using RNA-Seq-based measurement of immune infiltration (Supplementary Figure 8). Besides we observed that the RNA-ITH did not correlated with the purity of tumor samples. In contrast, we found that a significant correlation between the median RNA-ITH score and somatic copy-number alterations which are mainly caused by the tumor cells. This suggested that we could infer RNA-ITH based on the subclonal somatic SCNAs which were caused by the differences between tumor cell clones.

6. The authors state that: "We observed a significant correlation between the somatic mutations and the immune microenvironment (Spearman’s rho = 0.32, P = 1.1×10^{-13})." I think it is a bold statement when the correlation coefficient is only of 0.32. The *P* value does not say anything about the strenght of the correlation.

Response to comment: Thank you for the comments. We have updated the statement “We observed a weak significant but positive correlation between the somatic mutations and the immune microenvironment”. Besides, the spearman’s rho should be 0.35 which is marked in the Figure 4C and we have revised this number error in the sentence.

7. "The neoantigens from nonsynonymous mutations were used to explore neoantigen evasion. By implementing a bioinformatics pipeline to identify

neoantigens from the tumours (Supplementary Methods), a median of 107 predicted neoantigens per tumour (range from 42 to 754) was detected, and also neoantigen heterogeneity varied across the ESCC cohort." Potential neoantigens should be identified making use of the RNA analysis as well, thus, with only expressed mutations.

Response to comment: We do appreciate for your comments. In our study, neoantigen were defined as peptides with a rank percentage score $< 2\%$. We can also observed alternative neoantigen-depletion mechanism at the RNA level¹³. The repression of neoantigenic transcripts reflects the immune selection pressure. So we identified potential neoantigen making use of both expressed and not expressed mutations.

8. " the methylation level of genes with clonal not expressed neoantigen in the high TIL group is significantly higher than the clonal neoantigen in the low TIL group (Supplementary Fig. S10h, $P = 0.00439$)." Was this specifically encountered in genes carrying putative neoantigens? Meaning, when comparing the total of expressed genes vs non-expressed, I would also expect to see increased methylation in the latter.

Response to comment: We appreciated for your important suggestion. According to the reviewer's suggestion, the group of genes compared between H and L group is not same. We compared the methylation level of neoantigen promoters between different immune microenvironment and the neoantigen mutations between different patients are not same. After methylation level correct based on purity, we did not observe significant difference between the two groups ($P > 0.05$). This suggested the difference between the two groups are affected by multiple factors included purity or the 'baseline' methylation levels of the genes' promoter. After deep consideration, we decided to deleted this part and the conclusions which included Supplementary Figure 10F-H. Thanks again for pointing out the problem of our manuscript.

Reference

1. Hua X, *et al.* Genetic and epigenetic intratumor heterogeneity impacts prognosis of lung adenocarcinoma. *Nature communications* **11**, 2459 (2020).
2. Jimenez-Sanchez A, Cast O, Miller ML. Comprehensive Benchmarking and Integration of Tumor Microenvironment Cell Estimation Methods. *Cancer research* **79**, 6238-6246 (2019).
3. Bindea G, *et al.* Spatiotemporal dynamics of intratumoral immune cells reveal the immune landscape in human cancer. *Immunity* **39**, 782-795 (2013).
4. Davoli T, Uno H, Wooten EC, Elledge SJ. Tumor aneuploidy correlates with markers of immune evasion and with reduced response to immunotherapy. *Science* **355**, (2017).
5. Danaher P, *et al.* Gene expression markers of Tumor Infiltrating Leukocytes. *J Immunother Cancer* **5**, 18 (2017).
6. Newman AM, *et al.* Robust enumeration of cell subsets from tissue expression profiles. *Nature methods* **12**, 453-457 (2015).
7. Becht E, *et al.* Erratum to: Estimating the population abundance of tissue-infiltrating

- immune and stromal cell populations using gene expression. *Genome biology* **17**, 249 (2016).
8. Li B, *et al.* Comprehensive analyses of tumor immunity: implications for cancer immunotherapy. *Genome biology* **17**, 174 (2016).
 9. Aran D, Hu Z, Butte AJ. xCell: digitally portraying the tissue cellular heterogeneity landscape. *Genome biology* **18**, 220 (2017).
 10. Li S, *et al.* Dynamic evolution of clonal epialleles revealed by methclone. *Genome biology* **15**, 472 (2014).
 11. Chen X, *et al.* epihet for intra-tumoral epigenetic heterogeneity analysis and visualization. *Scientific Reports* **11**, 1-8 (2021).
 12. Favero F, *et al.* Sequenza: allele-specific copy number and mutation profiles from tumor sequencing data. *Annals of Oncology Official Journal of the European Society for Medical Oncology* **26**, 64 (2015).
 13. Biswas D, *et al.* A clonal expression biomarker associates with lung cancer mortality. *Nature medicine* **25**, 1540-1548 (2019).

14. Hou Y, *et al.* Single-cell triple omics sequencing reveals genetic, epigenetic, and transcriptomic heterogeneity in hepatocellular carcinomas. *Cell research* **26**, 304-319 (2016).
15. DNA methylation loss in late-replicating domains is linked to mitotic cell division. *Nature genetics*.
16. Li S, *et al.* Distinct evolution and dynamics of epigenetic and genetic heterogeneity in acute myeloid leukemia. *Nature medicine* **22**, 792-799 (2016).
17. M J-H, *et al.* Tracking the Evolution of Non-Small-Cell Lung Cancer. *The New England journal of medicine* **376**, 2109-2121 (2017).
18. Quinlan AR, Hall IM. BEDTools: a flexible suite of utilities for comparing genomic features. *Bioinformatics* **26**, 841-842 (2010).
19. Zhang M, *et al.* Clonal architecture in mesothelioma is prognostic and shapes the tumour microenvironment. *Nature communications* **12**, 1-12 (2021).
20. Masoodi T, *et al.* Evolution and Impact of Subclonal Mutations in Papillary Thyroid Cancer. *The American Journal of Human Genetics* **105**, 959-973 (2019).

21. Lee WC, *et al.* Multiregion gene expression profiling reveals heterogeneity in molecular subtypes and immunotherapy response signatures in lung cancer. *Modern pathology : an official journal of the United States and Canadian Academy of Pathology, Inc* **31**, 947-955 (2018).

REVIEWER COMMENTS

Reviewer #1 (Remarks to the Author):

I recognize that the authors tried to faithfully address the concerns raised in the previous review, but the revised version of the manuscript still has a few points that need further clarifications. Importantly, based on the reviewers' comments, the authors decided to delete some crucial arguments that were not strongly supported by their results, but due to that it became unclear how the study characterized the multi-omics trajectory of cancer evolution towards immune evasion. It seems that the study needs more concrete evidence on how the interactions between omics layers influences the evolution of HSCC under the immune microenvironment surrounding the tumors.

Major Comments

1. According to the manuscript, the authors argue that the transposition of L1 is associated with chromosomal instability in ESCC by showing the correlation between the methylation level of L1 and SCNA-ITH. However, in Supplementary Fig. S6G, the y-axis corresponds to L1 methylation ITH, but not its methylation level. It should be clarified whether the authors are emphasizing the 'absolute levels' of DNA methylation or 'ITH' of DNA methylation around L1 elements. Furthermore, the study is implicitly assuming that the DNA methylation levels (or ITH) of L1 corresponds to the amount of L1 transposition, even though the RNA expression of L1 elements is not directly associated with the DNA methylation levels (according to the authors' rebuttal). Are there any data supporting that the levels of L1 methylation (or ITH) is associated with the activity of L1 transposition? Moreover, it should be clarified in the main text whether the authors are claiming that the increased L1 transposition is one of the causes leading to CIN.

2. The title of the manuscript 'Tracking the evolution of esophageal squamous cell carcinoma under dynamic immune selection by multi-omics sequencing' as well as the section title 'Immune infiltration drives the heterogeneity across different omic level' imply that the analyses provide the multi-omics level interpretation of cancer evolution upon the immune microenvironment surrounding the tumor. However, the revised version of the manuscript only deals with genomic features, especially the loss-of-heterozygosity of HLAs, and it does not discuss about the transcriptomic and methylomic nature of cancer evolution immune evasion, which make it hardly arguable as a multi-omics interpretation of cancer evolution. Of note, the original version of manuscript addressed such challenge by claiming the suppression of neoantigen-producing genes by promoter hypermethylation, but it seems that the authors decided to delete that part in the revision since they thought the evidence are not strong enough to support their argument.

Minor Comments

1. (Line 202) Incomplete sentence: As LINE-1 is a well characterized phenomenon in cancer
2. (Line 204) '~ we found that there is significant correlation between L1 and SCNA-ITH'
3. (Line 1003~1012) Legends for Supplementary Fig. S10 do not seem to match with the corresponding figure panels. Please check panel (B), (D) and (E).

Reviewer #2 (Remarks to the Author):

I appreciate that the authors have addresses some of my comments. However, after reading the revised work, there are still outstanding and significant issues with this manuscript

- 1) I still think for the better understanding of the methylation ITH at the DMR level, a representative genomic window showing such methylation ITH would be helpful for the readers.
- 2) The paired copy number phylogenetic trees and phyloepigenetic trees from the two representative cases look incredibly similar, which makes me wonder the biological underpinning: both methylation and copy number changes are mostly stochastic and random, with only a minor fraction being relevant to cancer biology. Therefore, how do we understand this extreme level of similarity? The authors should discuss this. Moreover, the authors should also present cases where copy number phylogenetic trees and phyloepigenetic trees are not quite similar, so we have a whole picture of the interplay between genomic and DNA methylation ITH.
- 3) Regarding my previous comment on purity (#3), the authors pasted a plot without any legend and did not mention what it was for. I am guessing that it was meant to show RNA-ITH was not correlated with tumor purity? But I find it troubling that many tumors are so impure, with only 20%-30% of cancer cells. In fact, more than half of the tumors are estimated to have less than 30% tumor cells. None of the tumors had more than 60% cancer cells as the original manuscript described. Thus, the data quality is in question with such impure samples (none of the mathematical correction methods actually works really well, unless I am convinced with real data).
- 4) Relatedly, only RNA-ITH was shown to be not correlated with tumor purity. How about SCNA ITH and methylation ITH?
- 5) Regarding my previous comment #4, Fig.2E still carries little biological differences. There is almost no change between these groups although statistically significant. I am not sure any concrete conclusion can be drawn from here

Reviewer #3 (Remarks to the Author):

The authors have done a good job in the revision of their manuscript. No further comments.

Dear Reviewers:

Thank you very much for giving us an opportunity to revise our manuscript in this pandemic time. We appreciate reviewers very much for constructive comments and suggestions on our manuscript.

We do our best to point to point response to reviewers' comments. We have made a careful revision on the original manuscript. All revised portions are marked in capitalized word in the revised manuscript which we would like to submit for your kind consideration.

Kind regards.

Enver Tariq

Shixiu wu

2022.05.15

The point to point response as follows:

Reviewer #1

1. According to the manuscript, the authors argue that the transposition of L1 is associated with chromosomal instability in ESCC by showing the correlation between the methylation level of L1 and SCNA-ITH. However, in Supplementary Fig. S6G, the y-axis corresponds to L1 methylation ITH, but not its methylation level. It should be clarified whether the authors are emphasizing the 'absolute levels' of DNA methylation or 'ITH' of DNA methylation around L1 elements. Furthermore, the study is implicitly assuming that the DNA methylation levels (or ITH) of L1 corresponds to the amount of L1 transposition, even though the RNA expression of L1 elements is not directly associated with the DNA methylation levels (according to the authors' rebuttal). Are there any data supporting that the levels of L1 methylation (or ITH) is associated with the activity of L1 transposition? Moreover, it should be clarified in the main text whether the authors are claiming that the increased L1 transposition is one of the causes leading to CIN.

Response to comment: Thank you for the critical suggestion and for pointing out the problem with our manuscript. In Supplementary Fig. 6G, we used the ITH of DNA methylation (not the absolute DNA methylation) to correlate with the SCNA ITH. In the manuscript, we mistakenly use the term "L1 transposition" corresponding to the methylation level of LINE-1. Indeed, upon our ESCC dataset, we can't identify the amount of L1 transposition. However, we think this is quite an exciting direction; we could generate an omics dataset (whole genome sequencing data) of the ESCC cohort to validate the relationship in our subsequent study. We have revised our manuscript regarding the term used, making it more accurate.

2. The title of the manuscript 'Tracking the evolution of esophageal squamous cell carcinoma under dynamic immune selection by multi-omics sequencing' as well as the section title 'Immune infiltration drives the heterogeneity across different omic level' imply that the analyses provide the multi-omics level interpretation of cancer evolution upon the immune microenvironment surrounding the tumor. However, the revised version of the manuscript only deals with genomic features, especially the loss-of-heterozygosity of HLAs, and it does not discuss about the transcriptomic and methylomic nature of cancer evolution immune evasion, which make it hardly arguable as a multi-omics interpretation of cancer evolution. Of note, the original version of manuscript addressed such challenge by claiming the suppression of neoantigen-producing genes by promoter hypermethylation, but it seems that the authors decided to delete that part in the revision since they thought the evidence are not strong enough to support their argument.

Response to comment: Thank you for the careful reading and comments. According to the reviewer's suggestion, we further investigate the association between the tumor immune microenvironment and investigate the transcriptomic nature of immune evasion. Tumor cells have utilized several means to escape the TME recognition through HLA-related mechanisms. They can change the HLA expression to downregulate the expression of MHC complexes and therefore display fewer identifying antigens. The TME is an important factor which affected the HLA expression in ESCC. We assessed the relationship between HLA gene expression and infiltration of different immune cell types in ESCC. Based on the correlation patterns between immune cell infiltration and HLA and B2M expression, we found that HLA expression showed a significant positive association between HLA and B2M gene expression and the total immune infiltration, CD8 T cells, Cytotoxic cells and T cells which indicating high HLA gene expression is related with a relatively hot tumor microenvironment (Supplementary Fig. 11G). This result is consistent with previous TCGA research¹. We further investigate relationship between the ITH of HLA gene expression and the ITH of TME. We found that the ITH of CD8 T cells and NK cells are significantly correlated with the ITH of HLA-A, HLA-B and B2M expression (Figure 4D). This result strongly suggested that the transcriptomic ITH of HLA gene expression is driven by the immune infiltration.

Supplementary Fig. 11G Correlation heatmap between the HLA, B2M expression and tumor immune infiltration cells.

Figure 4D. Correlation heatmap between the ITH gene score of HLA, B2M and ITH score of tumor immune infiltration cells.

Reviewer #2

1. I still think for the better understanding of the methylation ITH at the DMR level, a representative genomic window showing such methylation ITH would be helpful for the readers.

Response to comment: Thank you for the reminding. According to the reviewer’s suggestion, we showed a representative genomic window to explain the methylation ITH which could help the readers understand the methylation ITH directly. The bar in blue represented the significant differential methylated regions (DMR) and the bar in grey represented the non-significant regions. We got all the DMR regions to evaluate the methylation ITH of each patient using the APITH method.

Supplementary Fig. 7A

2. The paired copy number phylogenetic trees and phyloepigenetic trees from the two representative cases look incredibly similar, which makes me wonder the biological underpinning: both methylation and copy number changes are mostly stochastic and random, with only a minor fraction being relevant to cancer biology. Therefore, how do we understand this extreme level of similarity? The authors should discuss this. Moreover, the authors should also present cases where copy number phylogenetic trees and phyloepigenetic trees are not quite similar, so we have a whole picture of the interplay between genomic and DNA methylation ITH.

Response to comment: We thank the reviewer for raising this important concern. As proven in our study, the loci got a significant increase of entropy in the SCNA regions. This means that the methylation level change trend will depend on the original methylation level. For example, if the loci located in a fully unmethylated region, the methylation level will increased (because of the entropy change) when the SCNA occurred in this region. These results revealed that intratumor methylation heterogeneity varied in different genomic segments which is associated with the chromosomal instability. To get a whole picture of the interplay between genomic and DNA methylation ITH, we exhibited the phylogenetic and phyloepigenetic trees to delineate the relationship among the tumor samples from the same patient. We observed that phylogenetic trees inferred from DNA methylation closely recapitulated the phylogenetic trees from SCNA, which is consistent with previous study. Besides, we also find several patients got different topologic structures (but with partial similarity) of phylogenetic (left in Supplementary Fig. 7B) and phyloepigenetic (right in Figure 4) trees (Figure 3). This suggested that there still potential mechanism which regulated the methylation ITH during the evolution of ESCC tumors.

Supplementary Fig. 7B

ESCC17:

ESCC17 methylation level dendrogram

ESCC17 copy number dendrogram

ESCC36:

ESCC36 methylation level dendrogram

ESCC36 copy number dendrogram

3. Regarding my previous comment on purity (#3), the authors pasted a plot without any legend and did not mention what it was for. I am guessing that it was meant to show RNA-ITH was not correlated with tumor purity? But I find it troubling that many tumors are so impure, with only 20%-30% of cancer cells. In fact, more than half of the tumors are estimated to have less than 30% tumor cells. None of the tumors had more

than 60% cancer cells as the original manuscript described. Thus, the data quality is in question with such impure samples (none of the mathematical correction methods actually works really well, unless I am convinced with real data).

Response to comment: We do appreciate your careful reading and comments. Sorry for the inconvenience of the plot. This plot showed the correlation between the mean value of the samples from each patient and the RNA-ITH value of each patient. I think this result obviously does not exhibit the truth of our data. To further show the reliability of the dataset, we plotted the violin plot, which represented the purity of the samples from our ESCC cohort from the Sequenza. The median purity value of our ESCC cohort is 67%, and 70% of samples' purity is above 50% upon the evaluation of Sequenza results. Obviously, the scatter plot largely reduces the values of purity of our cohort. We borrowed Dhruva's method², which also helped show there is no correlation between RNA-ITH and WES tumor purity (Extended Data Fig.9c in this paper). But it should be noted that the mean of the purity values from NSCLC patients is also enriched in a relatively low range², but the TRACERX team got a really strict filtration of samples³.

Figure 5

4. Relatedly, only RNA-ITH was shown to be not correlated with tumor purity. How about SCNA ITH and methylation ITH?

Response to comment: We appreciate the important suggestions. First, the SCNA and methylation level detection will be affected by the tumor cellularity composition and thereby affect the SCNA ITH and methylation ITH values. Second, we strictly picked the samples based on the results of H&E, which showed relatively reliability in the last response. In our study, purity is an important factor that we need to control in our correlation analysis. To consolidate the conclusion of the correlation test of our study, We also performed partial correlation to estimate Pearson (linear) correlation between two variables while controlling for one or more other variables (An R Package for a Fast Calculation to Semi-partial Correlation Coefficients.). This method has been validated in many studies. We computed the partial correlation controlling for tumor purity using the *pcor.test()* function from the R package *ppcor*.

5. Regarding my previous comment #4, Fig.2E still carries little biological differences. There is almost no change between these groups although statistically significant. I am not sure any concrete conclusion can be drawn from here

Response to comment: We appreciate this critical suggestion. After deep consideration, we agree with your suggestions. The copy number alteration will increase the entropy or PDR of the corresponding loci. However, the methylation level change trend will depend on the original methylation level. The methylation level of hypermethylated loci will decrease and vice versa.

The purity will also be another important factor that contributed to the methylation ITH, and we make our effort to correct the bias from purity. We have deleted this part from the manuscript. Thank you again for the critical suggestion.

Reference

1. Schaafsma E, Fugle CM, Wang X, Cheng C. Pan-cancer association of HLA gene expression with cancer prognosis and immunotherapy efficacy. *British journal of cancer* **125**, 422-432 (2021).
2. Rosenthal R, *et al.* Neoantigen-directed immune escape in lung cancer evolution. *Nature* **567**, 479-485 (2019).

3. M J-H, *et al.* Tracking the Evolution of Non-Small-Cell Lung Cancer. *The New England journal of medicine* **376**, 2109-2121 (2017).

REVIEWERS' COMMENTS

Reviewer #1 (Remarks to the Author):

The authors have adequately addressed all my comments.

Reviewer #2 (Remarks to the Author):

The authors have now addressed my concerns carefully and comprehensively. I think the manuscript has been significantly improved. This is an important resource and research in the field of ESCC genomics.